

# Using two-way nesting technique AGRIF with MARS3D V11.2 to improve hydrodynamics and estimate environmental indicators

Sébastien Petton[1], Valérie Garnier[2], Matthieu Caillaud[3], Laurent Debreu[4], Franck Dumas[5]

[1]Ifremer, Univ Brest, CNRS, IRD, LEMAR, 11 Presqu'île du Vivier, F-29840 Argenton, France
[2]Ifremer, Univ Brest, CNRS, IRD, LOPS, 1625 Route de Sainte-Anne, F-29280 Plouzané, France
[3]Ifremer, DYNECO, 1625 Route de Sainte-Anne, F-29280 Plouzané, France
[4]INRIA, Univ Grenoble Alpes, CNRS, LJK, F-38000 Grenoble, France
[5]SHOM / STM / REC, 13 Rue de Châtellier CS 92803, 29228 Brest CEDEX 2, France

*Correspondence to*: Sébastien Petton (sebastien.petton@ifremer.fr)

**Abstract.** In the ocean, meso / submesoscale structures and coastal processes are associated with fine scales. The simulation of such features thus requires the hydrodynamic equations to be solved at high-resolution (from a few hundred meters down to a few tens of meters). Therefore, local mesh refinement is a primary issue for regional and coastal modelling. As over structured grids, AGRIF (Adaptive Grid Refinement In Fortran) library is committed to tackle this challenge. It has been
implemented in MARS3D, which is a numerical model developed by Ifremer (the French research institute for the exploitation of the sea) for coastal environmental researches and studies. The present paper describes how the dedicated implementation preserves some essential principles (mass conservation, constant preserving…) along with the induced constraints. The use and the performance of this new tool are detailed over two configurations that illustrate the wide range of scales and resolutions typically targeted by coastal applications. The first one is based on multiple high-resolution (500m) grids that pave the coastal
ocean over thousands of kilometres, allowing a continuum between the regional and coastal scales. The second application is more local and has a finer resolution (50m). It targets a recurrent question for semi-enclosed bays: the renewal time indicator. Throughout these configurations, the paper intends at comparing the two-way nesting method with the traditional one-way approach and highlights how MARS3D-AGRIF tool proves to be an efficient way significantly improve the physical hydrodynamics and bring it biological issues.

## 1. Introduction

In the ocean, many observations have clearly shown that turbulence is ubiquitous and that flows are turbulent at all scales (Capet, 2015). It is also admitted that capturing the whole range of oceanic scales is far beyond the capabilities of any numerical model and surely for a long time to come. Even Large Eddy Simulation (LES) approach dedicated to solve the direct forward turbulent energy cascade is far from being reachable except for strict localized places. Therefore, it is necessary not only to
develop relevant and efficient parametrizations at the subgrid scale, but also to propose refinement capabilities in ocean models in order to focus the computational grid on key-locations. A key region can varies considerably based on geographic or



dynamical considerations. As the circulation is obviously tightly controlled by the coastline and more generally by the bathymetry, increasing the resolution of the grid may be essential to properly catch the coastal morphology (e.g., estuaries, cape, peninsulas, small bays and lagoons...). Alternatively, a key region may be an area where essential processes take place
that shape the circulation or a fate of a coastal discharge, even far offshore. The Strait of Gibraltar, where internal jump and consequently an internal solitary waves train are generated, is a perfect illustration, as these features propagate for hundreds of kilometers (Naranjo et al., 2014) until they break and reinforce mixing. Thus, the generating area must be addressed with both sufficient resolution and even locally adapted physics (here non-hydrostatic) to reproduce such structures. And then they must be accurately propagated, possibly outside the refined area.

Two strategies can be investigated (and are available with some degree of efficiency and accuracy) to tackle the spatial refinement in limited areas. The first one relies on unstructured grids together with finite volume or finite element discretizations: numerous models such as Delft-3D (Roelvink and van Banning, 1995), SLIM (Delandmeter et al., 2018), TELEMAC-3D (Janin et al., 1993) or T-UGOm (Piton et al., 2020) have shown for years their capability to cope with complex geometrical features such as river deltas or continental slope. In addition to the need to create efficient tools to manage the grid
building, this kind of approach has raised the classical issues related to continuously adaptive subgrid parametrization: how to manage both temporal and spatial refinements so that computational cost required by the finest grid cells does not spill over to the entire grid? An alternative method applies to structured grids and provides a refinement capability (adaptive or not) to integrate recursively a hierarchy of grids at different resolutions. This kind of approach was proposed in the early eighties (Berger et Oliger, 1984) and has already been implemented in either academic works (Debreu et al., 2012; Penven et al., 2006)
or for large-scale realistic applications (Biastoch et al., 2009; Marchesiello et al., 2011) to improve the resolution and hence the realism of the local dynamics.

Such an approach, the Adaptive Grid Refinement In Fortran (AGRIF) (Debreu et al., 2008), is based on domain decomposition and allows partial overlapping of the grids. It is thus well suited for coastal applications and is expected to facilitate the offshore continuum. Consequently, the AGRIF software has been introduced into a coastal numerical model, namely MARS3D (Lazure
and Dumas, 2008). Implemented along the coasts of the north western European shelf and over coastal bays, the mesh refinement should allow the representation of a large spectrum of spatial (small embayments and large areas) and temporal (fast as tides or and surges and slow as mesoscale or frontal instabilities) scales.

All along the coast, as the shallow ocean is subject to large environmental and anthropogenic pressures, the fate of coastal waters is a key for environmental, ecological and economic issues. Therefore, global and local indicators are crucial for
stakeholders to anticipate the spill of different types of materials such as oil (Jordi et al., 2006) or micro-plastics (Frère et al., 2017), biogeochemical processes due to nutrients flushing (Fiandrino et al., 2017; Le Pape and Menesguen, 1997), pollution phenomena (Jiang et al., 2017; Neal, 1966) or to develop restoration solutions (Kininmonth et al., 2010; Rossi et al., 2014; Thomas et al., 2014). These indicators must integrate complex hydrodynamics and highlight the most sensitive areas with respect to any specific ecological issue. The selection of the most appropriate indicator depends directly on the geographic
context and the hydrologic forcings of the studied area. A prior knowledge of the dynamics of the area is thus undoubtedly



necessary as well as an accurate numerical tool to monitor the ocean coastal circulation and study the relevance of the indicators.

This paper aims to demonstrate the capabilities and benefits of the two-way nesting method for the modelling of the interplay of coastal and regional dynamics. Section 2 briefly describes the hydrodynamic model MARS3D, the way AGRIF is

implemented and introduces shortly the time scale indicators computed. The two (regional and coastal) modelized configurations are then depicted with a general description of the in situ data. In Sect. 3, the running performance are firstly addressed. The boundary impact and the validation results are then given for both configurations. Eventually, a focus is made on the time scale indicators estimated into the coastal configuration. Finally in Sect. 4, these results are discussed regarding the comparison between one-way *vs* two-way nesting methods.

## 2. Material and Methods

### 2.1. The MARS3D model (V11.2 released in March 2018)

The MARS3D (Model for Applications at Regional Scale) model solves the primitive equations, which are obtained from the Navier Stokes equations expressing the main physical principle of mass and momentum conservation, under the main following assumptions:

• the density only slightly departs from a mean value $\rho_0$, namely the Boussinesq assumption,

• the horizontal scale of the considered oceanic flows is at least one order of magnitude larger than the vertical scale, which leads to the hydrostatic assumption.

MARS3D uses finite differences over a staggered C-grid in Arakawa classification (Fig. 1). This grid allows to enhance the effective resolution for a given space discretization $\Delta x$ and avoids the classical checkerboard effect inherent to the

colocalization of pressure (noted $q_{i,j}$ on Fig. 1) and the velocity points (noted $u_{i+\frac{1}{2},j}$ and $v_{i,j+\frac{1}{2}}$) as in the non-staggered A-grid.

MARS3D is fully described in Lazure and Dumas (2008) and is based on the methodology introduced by Blumberg and Mellor (1987). It relies on a mode (barotropic/baroclinic) splitting: the barotropic mode, which obeys the shallow water equation systems, is treated with a modified Alternating Direction Implicit (ADI) algorithm. The original ADI algorithm (Leendertse

and Gritton, 1971) has been modified to Eq. (1) in order to reduce the space splitting error. Equations (1) describe the first-time step during which $\eta$ (the surface elevation) and $u$ (the x-component of the barotropic velocity) are integrated together. The directional splitting error is reduced by performing a prediction for the y-component of the velocity noted here $v^{n+1,*}$. The second time step has a similar and symmetrical form with respect to $u$ and $v$.

$$v^{n+1,*} = v^n - g\Delta t \frac{\partial \eta^n}{\partial y} + \Delta t G_v(u^{n-1}, v^{n-1})$$

$$\eta^{n+1} = \eta^n - \Delta t \left[ \frac{\partial}{\partial x}(h^n u^{n+1}) + \frac{\partial}{\partial y}(h^n v^{n+1,*}) \right]$$



$$u^{n+1} = u^n - g\Delta t \frac{\partial}{\partial x}(\alpha \eta^n + (1-\alpha)\eta^{n+1}) + \Delta t G_u(u^n, v^n) \tag{1}$$

where $h$ is the total water depth, $\Delta t$ is the time step and $\alpha$ is the implicit coefficient of the external pressure gradient term. The term $G(u^n, v^n)$ gathers the vertical average of all the remaining terms including the non-linear and the dissipation terms, the Coriolis force, the friction at the surface and the bottom.

In a context of massive parallel computation, the system (1) is transformed into the strictly equivalent system (2) thanks to the splitting of $u^{n+1}$ into an explicit part and an implicit one. This system leads to unknown vectors made solely of $\eta^i$. Therefore, for a given j, the size is exactly the number of grid cells in the x-direction instead of twice this size for the regular ADI algorithm for which unknown vectors are made of $\eta^i$ and $u^i$. Thus, it reduces the size of the data involved in the message passing interface between CPUs by a factor of 2.

$$v^{n+1,*} = v^n - g\Delta t \frac{\partial \eta^n}{\partial y} + \Delta t G_v(u^{n-1}, v^{n-1})$$

$$u^{n+1,*} = u^n - g\Delta t\alpha \frac{\partial \eta^n}{\partial x} + \Delta t G_v(u^n, v^n)$$

$$-g\Delta t^2(1-\alpha)\frac{\partial}{\partial x}\left[h^n\frac{\partial \eta^{n+1}}{\partial x}\right] + \eta^{n+1} = \eta^n - \Delta t\left[\frac{\partial}{\partial x}(h^n u^{n+1,*}) + \frac{\partial}{\partial y}(h^n v^{n+1,*})\right]$$

$$u^{n+1} = u^{n+1,*} - g\Delta t(1-\alpha)\frac{\partial \eta^{n+1}}{\partial x} \tag{2}$$

As MARS3D mostly targets the coastal dynamics, it gets a drying and flooding capability which modifies the expression of
the original continuity equation form for partially dried cells into Eq. (3):

$$\eta^{n+1}Sf_{wet}^{n+1} = \eta^n Sf_{wet}^n - \Delta t\left[\partial_x(h^n u^{n+1})\Delta y + \partial_y(h^n v^{n+1})\Delta x\right] \tag{3}$$

where S stands for the grid cell surface ($S = \Delta x \Delta y$) and $f_{wet}$ is the time-dependent fraction of the grid cell which is flooded ($0 \le f_{wet_{i,j}} \le 1$). $f_{wet}$ is given by a prognostic equation accounting for the bottom slope and the sea surface elevation gradient and $0 \le f_{wet_{i,j}} \le 1$.

Last but not least the barotropic / baroclinic coupling is rather straightforward because both modes are integrated with the same time step; the coupling mainly consists in correcting the 3D currents in order that their vertical average matches the barotropic current.

**2.2. The AGRIF model: Mesh refinement and grid interactions**

AGRIF (Adaptive Grid Refinement In Fortran from Debreu et al. (2008)) is a package for the integration of Structured
Adaptive Mesh Refinement (SAMR) features within a multidimensional finite difference model. Its main objective is to simplify the integration of SAMR potentialities within an existing model, whilst making minimal changes. In particular, it includes a lexicographic analyser of Fortran code that generates, at the compilation step, the data structures required for running the same code on any grid hierarchy. In addition to MARS3D (Dufois et al., 2014; Muller et al., 2009), AGRIF is currently





used in the following ocean models: ROMS-AGRIF (Debreu et al., 2012; Penven et al., 2006) a regional model developed
jointly at Rutgers and UCLA universities; NEMO ocean modelling system (Biastoch et al., 2009, 2018) a general circulation
model used by the European scientific and operational communities; and HYCOM a regional model developed jointly by the
University of Miami and the French Navy.

### 2.2.1. General algorithm

For a general review of two-way nesting algorithms, the reader is referred to Debreu and Blayo (2008). This section recalls
the general algorithm introduced in MARS3D. For the sake of simplicity, one considers a single child grid covering a
subdomain $\omega$ of the parent domain $\Omega$, as illustrated in Fig. 2. The boundary of the child grid is delimited by the interface $\Gamma$.
The coarse resolution grid has a mesh size given by $\Delta x_H$, while the fine resolution grid has a mesh size $\Delta x_h = \Delta x_H/\rho$ where
$\rho$ is the spatial mesh refinement ratio (it is an integer). The partial differential equations solved by the model are written in the
following form:

$$\frac{\partial q}{\partial t} = L(q)$$


from an initial condition and with lateral boundary conditions prescribed at the limits of $\Omega$. These equations are discretized
on the coarse and fine grid domains by:

$$\frac{\partial q_H}{\partial t} = L_H(q_H), \qquad \frac{\partial q_h}{\partial t} = L_h(q_h) \tag{4}$$

Thus $L_H$ and $L_h$ are the discretizations of the same continuous operator $L$ at different resolutions. The child grid requires lateral
boundary conditions at the interface $\Gamma$. In two-way mode, the coarse solution is updated using the fine solution. This is
modelled by two different operators: a spatio-temporal interpolator $(P)$ and a restriction operator $(R)$ respectively. Assuming
that the model is fully explicit, the algorithm can be written in the following simplified form:

1)      $q_H^{n+1} = L_H(q_H^n)$

2)      For $m = 1 \dots \rho_t$ do

$$q^{n+\frac{m}{\rho_t}}\Big|_\Gamma = P(q_H^n, q_H^{n+1})$$

$$q^{n+\frac{m}{\rho_t}} = L_h\left(q_h^{n+\frac{(m-1)}{\rho_t}}\right)$$

3)      $q^{n+1}|_\omega = R(q_h^{n+1})$

Here, $\rho_t$ is the time refinement factor ($\rho_t = \Delta t_H/\Delta t_h$) which is equal to the space refinement factor $\rho$ if the model is restricted
to a CFL (Courant Friedrichs Levy) stability condition. The step (1) corresponds to the integration of the coarse grid model
for one time step $\Delta t_H$ on $\Omega$, while the step (2) corresponds to the integration of the fine grid model over $\rho_t$ time steps. The
interpolator $P$ makes use of $q_H^n$ and $q_H^{n+1}$ to produce space and time interpolations on the interface $\Gamma$.



As previously mentioned, MARS3D model has a semi-implicit free surface formulation for the computation of the fast external gravity waves. On the fine grid, the simple approach described in Sect. 2.1 is integrated and the free surface at the boundary is interpolated from the coarse grid solution:


$$A_H \eta_H^{n+1} = \text{RHS}_H \tag{5}$$

$$A_h \eta_h^{n+\frac{m}{\rho_t}} = \text{RHS}_h, \qquad \eta_h^{n+\frac{m}{\rho_t}}\bigg|_\Gamma = P(\eta_H^n, \eta_H^{n+1}) \tag{6}$$

where $A_H$, $A_h$ are the discretizations of the Helmholtz equation (2) on the coarse and fine grids and $\text{RHS}_H$, $\text{RHS}_h$ the corresponding right-hand sides. This procedure introduces a loss of accuracy because errors produced by the coarse grid resolution (5) propagate inside the high-resolution domain through the boundary forcing in step (6). However, the update

schemes described in next subsection ensure that the barotropic component of the right-hand side on the coarse grid $\text{RHS}_H$ is strictly identical to the update of its equivalent in the fine grid. More complex iterative strategies, using a multigrid algorithm of the elliptic system, can be used to achieve the desired accuracy (e.g. Haley and Lermusiaux (2010), Martin and Cartwright (1996)). Still, the additional cost is significant, especially with the addition of the time refinement since in that case the coarse solver (5) has also to be solved at intermediate (fine grid) times.

**2.2.2. Free surface, tracer and velocity updates with wetting and drying**

In a free surface ocean model, for conservation reasons, the discrete time evolution of the free surface elevation can be written in terms of the divergence of a barotropic transport U and V in the x and y directions (volumetric fluxes):

$$S_{i,j} f_{wet_{i,j}}^{n+1} \eta_{i,j}^{n+1} = S_{i,j} f_{wet_{i,j}}^n \eta_{i,j}^n - \Delta t \left[ U_{i+\frac{1}{2},j} - U_{i-\frac{1}{2},j} + V_{i,j+\frac{1}{2}} - V_{i,j-\frac{1}{2}} \right] \tag{7}$$

A consistent update scheme for free surface and barotropic transport has be obtained by applying the restriction operator to the

right-hand side of this equation. Hereafter, one considers the situation represented in Fig. 3 where the mesh refinement coefficient is equal to 3. The free surface restriction operator is a simple average of the 9 fine grid cells (and assuming there is no time refinement) using the following area weighted formulae:

$$S_{i_c,j_c} f_{wet_{i_c,j_c}}^{n+1} \eta_{i_c,j_c}^{n+1} = \sum_{\substack{i=i_f-1,i_f+1 \\ j=j_f-1,j_f+1}} S_{i,j} f_{wet_{i,j}}^{n+1} \eta_{i,j}^{n+1} \tag{8}$$

where $i_c$ and $j_c$ are the indices of the cell in the coarse grid and $i_f$ and $j_f$ in the fine grid (see Fig. 3). Using (7), the time

evolution of the updated free surface is given by:

$$S_{i_c,j_c} f_{wet_{i_c,j_c}}^{n+1} \eta_{i_c,j_c}^{n+1} = S_{i_c,j_c} f_{wet_{i_c,j_c}}^n \eta_{i_c,j_c}^n - \Delta t \left[ \begin{array}{l} \left( U_{i_f+\frac{3}{2},j_f-1} + U_{i_f+\frac{3}{2},j_f} + U_{i_f+\frac{3}{2},j_f+1} \right) - \left( U_{i_f-\frac{3}{2},j_f-1} + U_{i_f-\frac{3}{2},j_f} + U_{i_f-\frac{3}{2},j_f+1} \right) \\ + \left( V_{i_f-1,j_f+\frac{3}{2}} + V_{i_f,j_f+\frac{3}{2}} + V_{i_f+1,j_f+\frac{3}{2}} \right) - \left( V_{i_f-1,j_f-\frac{3}{2}} + V_{i_f,j_f-\frac{3}{2}} + V_{i_f+1,j_f-\frac{3}{2}} \right) \end{array} \right] \tag{9}$$

Consistently with the average restriction operator for the free surface, the coarse grid barotropic transports can then be updated by the relations:





$$U_{i_c+\frac{1}{2},j_c} = U_{i_f+\frac{3}{2},j_f-1} + U_{i_f+\frac{3}{2},j_f} + U_{i_f+\frac{3}{2},j_f+1}$$

$$V_{i_c,j_c+\frac{1}{2},} = V_{i_f-1,j_f+\frac{3}{2},} + V_{i_f,j_f+\frac{3}{2}} + V_{i_f+1,j_f+\frac{3}{2}} \tag{10}$$

This corresponds for $U$ to an injection in the x−direction and an average in the y−direction and reciprocally for $V$. If time refinement is applied, these fluxes have been summed up during the $\rho_t$ fine grid time steps.

Another crucial point is that the free surface update using (8) must preserve constancy: if the fine grid free surface $\eta$ is spatially constant, the update operator has to preserve this constant. In MARS3D, this is achieved by updating the wet fractions of the coarse grid according to:

$$f_{wet_{i_c,j_c}}^{n+1} = \frac{1}{S_{i_c,j_c}} \sum_{\substack{i=i_f-1,i_f+1 \\ j=j_f-1,j_f+1}} S_{i,j} f_{wet_{i,j}}^{n+1} \tag{11}$$

Moreover, in order to keep constancy preservation, tracer values must be updated with the same update operator as for the free surface. The three-dimensional velocities (or more precisely volumetric fluxes) must also be updated with the same update operator as for the barotropic velocities.

### 2.2.3. Parallelization option

The MARS3D model and the AGRIF library can be run with either sequential mode, OMP mode, MPI mode or hybrid mode (based on both MPI and OMP parallelization). In the case of one unique child grid, both mother and child grids are resolved sequentially. In the case of multiple child grids, the user can choose between two options for the integration of the child grids: The first solution consists in a sequential integration with the same number of MPI processors for each child grid; The second solution involves a distribution of the MPI processors between each child grid. For this latter, the processors are allocated to the child grids at the beginning of the simulation, based on the number of wetting cells in every child grid and so that all child grids are integrated simultaneously. This last option is more benefic when the size of child grids differs from one another or when the child grids are much smaller than the mother grid.

### 2.2.4. Communication between the grid hierarchy

At the boundaries of the subdomains, the mother grid provides the high-resolution grids with the free surface, tracers and velocity components. These variables are interpolated onto the fine grid with Piecewise Parabolic Method. In addition, a sponge layer is set along the open boundaries. As explained in Debreu and Blayo (2008), the sponge layer is implemented as a diffusion term that acts on the difference between the high-resolution solution and the interpolation of the coarse resolution solution on the fine grid:

$$\frac{\partial q_h}{\partial t} = L_h + \nabla \cdot \left( \mu \nabla (q_h - I_H^h q_H) \right)$$



where $\mu$ is a coefficient ranging from its maximal value $\mu_0$ at the interface to 0 a few grid points away from it (usually at a distance of 3 coarse grid cells). This sponge layer is applied both on momentum and tracers.

After the time integration of the high-resolution grids, the information is fed back to the parent grid in the two-way context: the updated coarse solution becomes the spatial average of the fine solution. In order for this restriction operator (R) to keep

the fluxes coherent and conserve mass, the mother and child bathymetries have been constructed so that the bathymetry reduction conserves volume. When some grids at the same level of the hierarchy overlap, their fields are weighted before applying the reduction. The weights are estimated once and for all during the pre-processing phase and are intended, for each sub-cell of a coarse cell, to favor(/disfavor) a child grid over another if its sub-cell is further from (/closer to) the open boundary. This functionality is only available with one zoom level so far. This constraint will be taken over with the next upgrade of

AGRIF library and will be addressed in the library itself to simplify AGRUF portability into numerical models.

### 2.3. Two configurations: a regional and a coastal cases

Two configurations have been implemented with MARS3D/AGRIF to show its capabilities both on hydrodynamics and on tracer applications: One is focused geographically while the other one covers a large region. They both offer a wide span of applications.

#### 2.3.1. Bay of Brest configuration

The Bay of Brest is a typical coastal region of prime interest for human activities (such as transport, aquaculture, fishing and leisure activities). This semi-enclosed macrotidal ecosystem is located at the western end of Brittany (France), spanning over 180 km². It is connected to the Iroise Sea through a 1.8-km wide by 6-km long and roughly 50-m deep inlet (called the Goulet de Brest). The strong hydrodynamic currents due to its complex geometry and topography are relatively well known, being

the purpose of many previous studies. This macro-tidal coastal area is characterised by a dominant semi-diurnal tide with a tidal range of 1.2 to 7.3 m. The tidal currents peaks up to 3 m s$^{-1}$ in the Goulet and are in quadrature phase relative to the surface elevation (Petton et al., 2020). The mean volume at mid-tide inside the bay is roughly 2 billion m3. As its average depth is only of 8 m, the back-and-forth flow at each tide prevents stratification nearly everywhere (Le Pape and Menesguen, 1997). The tidal prism is 25% of the mean volume in neap tide and 60% in spring tide. The hydrology is dominated by

freshwater runoffs coming mostly from the Aulne river (Auffret, 1983). The atmospheric forcings rely on the Météo France AROME (2.5 km) analysis with hourly data.

The MARS3D/AGRIF model is set up over the Iroise sea (geographic limits 47.74°N - 48.82°N and 4.08°W - 5.55°W) with a horizontal grid resolution of 250m. A zoom over the Bay of Brest is introduced with a resolution of 50m (geographic limits 48.20°N - 48.44°N and 4.09°W - 4.72°W), see Fig. 4. The time and space refinement factors are both equal to 5. The vertical

discretization is performed with 20 equidistant $\sigma$-layers in both grids. As the previous configuration, the bathymetries are interpolated from a combination of different digital terrain models (SHOM, Ifremer, IGN). The Iroise model is forced by harmonic components from the SHOM CST-France model (Le Roy and Simon, 2003). The 3D open boundary conditions for



baroclinic currents, temperature and salinity are imposed at hourly frequency from the previous regional configuration based on a hindcast (Caillaud et al., 2016). Freshwater inputs for the four main rivers in the Bay have been taken from the French

HYDRO database (http://www.hydro.eaufrance.fr/) and corrected with corresponding watershed surface rates.

The MARS3D/AGRIF Bay of Brest configuration is available with a more detailed description of physical, numerical and parallelisation parametrisations from Petton and Dumas (2022).

### 2.3.2. Bay of Biscay configuration

The MARS3D model has already been used to investigate the Bay of Biscay and its extension to the western English Channel

(Huret et al., 2013; Lazure et al., 2009). Here, the MARS3D/AGRIF capability is implemented along the North-Western European continental shelf. Figure 5 shows how the AGRIF skill is used to pave the coast line from Spain to Belgium with barely overlapping grids. Seven zooms with a resolution of 500m and roughly the same grid size are introduced into a coarser 2.5 km resolution grid that encompasses all the seven grids. The coarser grid is itself included in a larger 2D model (5km resolution) in which the tide is imposed using the FES 2014 ocean tide atlas with 14 harmonics constituents (Lyard et al.,

2021). The 3D open boundary conditions for temperature and salinity of the coarser grid are provided at a daily frequency with the MERCATOR PSY2V4 re-analysis model. The main hydrological runoffs are set in the zooms (96 rivers accounted for) from different source databases (Spain, English, French and Netherlands). The atmospheric forcings are picked up in the Météo France ARPEGE High Resolution (0.1°) analysis with hourly data for the coarser grids. For the child grids, the atmospheric forcings rely on the Météo France AROME (2.5 km) analysis with hourly data. The discretisation over the vertical axis is

performed with 40 generalized $\sigma$-layers with a stretching function that induces refinement above 150m depth, next to the surface. A space refinement factor of 5 is used between each grid level but the time refinement is adapted according to the maximum velocity encountered locally. Thus, the time refinement coefficient is heterogenous: it is either 3 (over areas with rather slow flows) or 5 (in very energetic areas such as in the middle of the English Channel). Over the overlapping areas at zoom level, a distance-weighted interpolation is used to combine information from one child grid to another.

### 2.3.3. One-way configurations

Both previous configurations have also been set with classical offline one-way nested configuration. Among the seven child grids within the regional configuration (Bay of Biscay), the Iroise child grid has been separately deployed with one-way nesting procedure for the purpose of this paper. First, the coarser model has been run independently over the whole computation period to get information to force the child grid area at hourly frequency. There are baroclinic currents, temperature and salinity

variables. Those fields are projected onto the child grid with an external homemade Fortran tool to get classical open boundary conditions. Thanks to the proximity of the child grid border to French coast, the available harmonic components from the SHOM CST-France model have been used to predict the tidal sea surface elevation at the open boundaries. The other forcings such as atmospheric conditions or runoffs are exactly the same than the ones used in the two-way simulations.



## 2.4. Timescale indicator applied to the coastal configuration

There are numerous indicators of hydrological characteristics based on theories of transport timescales. Over the years, many studies (Bolin and Rodhe, 1973; Monsen et al., 2002; Takeoka, 1984; Zimmerman, 1976) have defined and assessed different scales to describe water renewal on a particular spatial scale (a bay, an estuary, a harbour) in which mixing processes will renew the water mass (through open-ocean–connected boundaries or forced by runoff inputs). However, the vocabulary remains very diverse (Bacher et al., 2016). Those time scales are often provided in the framework of the constituent-oriented

age and residence time theory (CART, www.climate.be/cart), under different names depending on their exact purpose (de Brauwere et al., 2011; Deleersnijder et al., 2001; Delhez, 2006; Delhez et al., 2004). They can also be used to specify the transport in the vertical direction (Bendtsen et al., 2009; Meier, 2005; White, 2007). Lucas and Deleersnijder (2020) have made a specific review of the whole set of indicators estimated either with Eulerian method or Lagrangian computation with particle-tracking. Most of these timescales can be averaged over the whole basin or can be defined locally, at every position in the

basin, to provide a more detailed spatially distributed information on the water renewal capacity of a basin (Jouon et al., 2006). Moreover, under certain physical circumstances (large diffusion processes and weak runoffs), they can be identical (Viero and Defina, 2016b).

Our intent is not to debate the pros and cons of the different indicators that exist and the way they are computed. Here, the goal is to characterize the renewal process of the Bay of Brest (coastal configuration) that can be summarized as a semi-enclosed

area with large advection movements (tidal mixing) and relatively weak continental runoffs. Therefore, the *e*-folding flushing time is evaluated at the global and local spatial scales, which requires accurate consideration of the return flow due to the tide back-and-forth movements. This water renewal time indicator is a typical example to highlight the two-way nesting capabilities of MARS3D/AGRIF.

### 2.4.1. *e*-folding flushing time

The water renewal time is based on an eulerian reference system. It estimates the dilution time scale of a passive tracer released within a control domain. It indicates the time spent for exchange water masses with new water coming from both the sea and the rivers discharging into the basin. In our case, the control domain is made of the whole inside volume of the bay plus the external part link to its tidal prism computed in mean tide conditions. The return flow through the Goulet due to the semi-diurnal tide must be accounted for in the estimation of the renewal time scales. As recommended by Viero and Defina (2016),

the simulated domain must be really larger than the control volume. This can be done by using the two-way nesting and update skills of AGRIF. The initial concentration of a passive tracer is set to 1 inside the control domain (in the fine grid) and 0 elsewhere. After the release, only waters coming from the ocean (outside the Iroise sea) and the rivers runoffs are fixed with a tracer concentration equal to 0. The local flushing time $\theta$ is estimated in each grid cells as the time decrease in concentration between 95% and $1/e$ of initial concentration using an exponential regression $C(t - t_0) = e^{-(t-t_0)/\theta}$ (Grifoll et al., 2013;

Jouon et al., 2006; Plus et al., 2009). The flushing lag $t_0$ enables to understand the evolution of water coming from outside or





river within the control volume. It enables a better regression for areas far away from the volume's boundaries. A global flushing indicator is also estimated for the whole control volume using the same method without the lag dependency. To get rid of the initial tide conditions, the 13 tracers are separately released every hour for an even coverage of the tidal cycle. This initial tide effect may be really important in estuary and will be discussed later on.

**2.4.2. Case scenario**

For the regional Bay of Biscay configuration, two realistic hindcasts have been realized over several years (2010-2020 for the two-way and 2016-2019 for the one-way) to demonstrate and to characterize nesting benefits. For each hindcast, a spin-up of 3 months has been performed. These two common years have been selected upon the available datasets for several validation parameters (tide gauge, temperature and salinity data).

For the coastal Bay of Brest configuration, two realistic hindcasts have been realized over 2017 up to 2019 for both one-way and two-way nesting techniques. As Brest is located at mid-latitude, the bay is subjected to highly variable and sometimes intense meteorological forcings. Therefore, various numerical experiments have been performed to catch these conditions and estimate a realistic renewal indicator. Each indicator simulation has been carried out according to the same protocol with a hydrodynamic spin-up run performed over one month before the release of passive tracers.

To obtain various tidal regimes and hydrologic runoffs, the study focuses on four scenarios with respect to the tidal range combined with two different runoffs: Releases have been done at the beginning of spring tides and neap tides, in winter and summer seasons during a flood and a low-flow events. All of these simulations have been performed with realistic atmospheric forcings. A detailed description of the scenarios is given in Table 1.

**Table 1: Environmental condition for each computation periods mean over the first 30 days of simulation**

| Scenario | Initial dates of modelling | Cumulated Aulne river flow ($10^6$ m$^3$) | Wind velocity ± SD (m.s$^{-1}$) and direction (°) |
|---|---|---|---|
| Low water – Neap tide | Jan 29th 2016 | 380.3 | 8.43 ± 4.0 (257°) |
| Low water – Spring tide | Feb 6th 2016 | 337.7 | 8.29 ± 4.2 (274°) |
| Flood – Neap tide | Aug 8th 2015 | 13.7 | 4.69 ± 2.6 (281°) |
| Flood – Spring tide | Jul 29th 2015 | 10.5 | 4.97 ± 2.7 (242°) |


**2.5. Data**

**2.5.1. Time series**

In this paper, the quality of the Iroise sea and the bay of Brest simulations are carefully assessed by comparison to data. Several environmental datasets are available thanks to long-term in situ monitoring programs. Three datasets are available in the



framework of the national program COAST-HF (Coastal Ocean Observing System-High Frequency, www.coast-hf.fr) which gathers fourteen automated moored buoys. The COAST-HF ASTAN buoy (48.749°N; 3.961°W) is a cardinal buoy of opportunity located 3.1 km offshore from Roscoff, east of the Batz Island. It records data every 30 minutes at 5-meter depth since 2008 (Gac et al., 2020), over a mean bathymetry of 45 m. The COAST-HF MAREL-Iroise buoy (48.357°N, 4.582°W) is located at the entrance of the bay of Brest and records data every 20 minutes at 2-meter depth since 2000 (Rimmelin-Maury
et al., 2020). Inside the bay of Brest next to the Mignonne river mouth, the COAST-HF SMART-Daoulas buoy (48.317°N, 4.331°W) is monitoring parameters at 50 cm over the seabed at 15-mins frequency since 2016 (Petton et al., 2021b). Next to this last point, the Ifremer observatory network ECOSCOPA has a study site called Pointe du Château (48.335°N, 4.319°W) on an oyster farm in the intertidal zone. Temperature and salinity data are available at a 15-mins frequency since 2008 (Petton et al., 2021a). We also had access to the sea surface temperature data from the Datawell buoy of les Pierres Noires which is
part of the swell monitoring network CANDHIS (CEREMA) and located in the middle of the Iroise Sea (48.29°N, 4.97°W). These monitoring stations are presented in Fig. 4 and Fig. 5.

### 2.5.2. Satellite data

Besides, satellite data are used for sea surface temperature validation at two different horizontal scales: The first one is based on SST fields extracted from the global Advanced Very High Resolution Radiometer (AVHRR) Pathfinder V5 daily dataset.
The ODYSSEA chain has been modified by Saulquin and Gohin, (2010) to use optimal interpolation for the reconstruction of gap-free and using the previous analysis as a first guess. The product is gridded at a 0.02° spatial resolution and freely available at https://resources.marine.copernicus.eu (Autret and Piollé, 2018). The second one is based on the Thermal InfraRed Sensor (TIRS) from the Landsat 8 satellite. As it orbits the Earth in a sun-synchronous, near-polar orbit inclined at 98.2 degrees, one gets a track over our area of interest (Iroise Sea in paragraph 3.2.2.) every 8 days. Consequently, it is hard to extract snapshots
without too much clouds. Recently the United States Geological Survey (USGS) has started to distribute Landsat Collection 2 Level 2 (values are given after atmospheric corrections) with a calibrated land surface temperature field. The development of a water temperature algorithm is not the aim of this paper and represents a challenge by itself (Vanhellemont, 2020). However, the use of such high-resolution product (30m gridded) is very useful to detect fine structures. In that respect, the ODDYSEA product is complementary to the Landsat 8 scene and a reference on a coarser grid. To discriminate water temperature from
cloud or land value, the quality index given for each pixel for this collection is used.

## 3. Results

### 3.1. Technical aspects

In classic one-way nesting, an important step to consider is the building of the set of fields prescribed at the open boundaries of the finest grid. This step is done offline from the output files of the coarser model. For the present study, the sea surface
height, baroclinic currents, temperature and salinity fields are of primary importance for the driven circulation so they have to



be interpolated on the fine grid, according to the chosen open boundary scheme. For both regional and coastal configurations, 3D output files are extracted at an hourly-frequency. This frequency has been chosen as a compromise between the disk space availability and the impact over the resolution of the hydrodynamics. The interpolation is made with an external homemade Fortran tool on a sequential node. As both coarse and fine MARS3D configurations use sigma vertical coordinates, the user

has to define an intermediate z-profile on which the coarse model fields are interpolated before performing the geographic interpolation. Depending on the number of z-levels and the desired vertical refinement, the processing can be computationally expensive. After this step, the fine model can be launched separately. With the AGRIF library, the two-way nesting enables interpolation from the coarse grid at each time step of the fine grid. This is done directly online and expert users can modify the interpolation scheme for each state variables if required..

The computational times for the simulation of 45 days are provided in the Table 2 for both MARS3D one-way nesting or MARS3D/AGRIF two-way nesting. All the simulations have been performed on the supercomputer DATARMOR infrastructure (https://wwz.ifremer.fr/pcdm). Since the computing performance depends on the load of the machine, the computational coast has been evaluated from a pool of repeated experiments. For each configuration, the different models (one-way and two-way) have been run with the same parallelized discretization but with two different parallelization methods:

a classic MPI domain decomposition and a hybrid computation based on both MPI (domain decomposition) and OMP (distribution of threads in Fortran loops inside each domain) parallelization. The hybrid way is generally recommended and this is even more essential for the use of AGRIF two-way nesting, because each grid has to be run on the same total number of processors. Besides in MARS3D, an optimized domain decomposition is made to balance the load on the different nodes of the cluster. This consists in attributing to each MPI core approximately the same number of wet grid cells and excluding the

land-masked part of the domain.

**Table 2: Mean computation time given in hours for both modelized configurations for a simulation of 45 days.**

| Configuration | Mode | Nodes | Scenario | one-way | two-way |
|---|---|---|---|---|---|
| Regional | MPI | 56 MPI nodes | Hydrodynamic only | 12.0 | 15.0 |
| Coastal | MPI | 112 MPI nodes | Hydrodynamic only | 13.5 | 18,6 |
| Regional | Hybrid | 56 MPI nodes with 8 OMP each | Hydrodynamic only | 2.9 | 4.3 |
| Coastal | Hybrid | 48 MPI nodes with 7 OMP each | Hydrodynamic & 13 tracers | 21.1 | 29.2 |

For both configurations, the computation time is increased by one third for the AGRIF configuration compared to the classic one-way nesting. This difference is explained with the constant spatial interpolation performed within the AGRIF library at

each fine grid time step. The update process which assures the fluxes continuity, covers the fine grid area in the coarse grid and is also responsible for this increase. For the one-way nesting chain, the handling time of 3D output files and the waiting





time due to DATARMOR load activity have not been taking into account. Their introduction would decrease the whole difference especially if several zooms are implemented.

## 3.2. Hydrodynamic impact over open boundary conditions

The ongoing issue for nesting models relies on open boundary conditions. For the AGRIF configuration, the tidal propagation is performed using sea surface elevation and fluxes interpolation. For the regional Bay of Biscay configuration, the initial tide forcing is imposed at the mother grid's boundary with FES model. It is composed of 12 tidal harmonic components whereas SHOM CST-France model used to force the child grid in one-way nesting contains 112 components. Moreover, this latter model has been accurately validated throughout the French tidal gauge network RONIM.

The modelled tidal signal is compared to hourly validated data from four tidal gauge available in the studied area. The main statistics are estimated over one year are given in the Table 3 for the regional Bay of Biscay configuration. The tide from the two-way techniques is slightly less precise than the one-way simulation. This occurs especially at the north-eastern boundary, at Roscoff, where the RMSE is twice as large. On the opposite at the southern boundary, at Concarneau, the differences are in favour of the two-way nesting. The same validation has been done for the coastal Bay of Brest configuration with the Brest

tidal gauge. The results between one-way and two-way techniques are similar (nstd = 1.009 / rmse = 9.4 cm / pcc = 0.998).

**Table 3: Comparison of sea surface elevation between MARS3D one-way (normal font) and MARS3D/AGRIF two-way (bold font) for the Iroise zoom (500m horizontal resolution) compare to RONIM tidal gauges.**

|  | Roscoff | Le Conquet | Concarneau | Brest |
|---|---|---|---|---|
| **Normalize standard deviation (%)** | 1.022 | 1.010 | 1.029 | 0.988 |
|  | **0.969** | **1.004** | **1.024** | **0.990** |
| **Root mean square error (cm)** | 9.3 | 7.2 | 7.9 | 10.5 |
|  | **18.7** | **9.4** | **6.4** | **12.2** |
| **Pearson correlation coefficient (%)** | 0.999 | 0.999 | 0.998 | 0.998 |
|  | **0.985** | **0.993** | **0.995** | **0.990** |

To observe the influence of the tide propagation, the two nested configurations have been compared to the PREVIMER

harmonic component atlas with the zoom grids. This atlas has been built upon a series of barotropic simulations at 250m horizontal resolution and validated with the RONIM network (Pineau-Guillou, 2013). A harmonic decomposition has been applied to each hindcast with an hourly output frequency over a period of one year. In Table 4, a comparison of wave elevations is given for the waves M2, S2, O1, K1 at 4 different points distributed in the Iroise sea. Both models are in rather good agreement with the reference atlas. The main difference is found for the MARS3D/AGRIF two-way configuration for the K1

wave on relative difference, where the wave errors are over estimated by 20%, but the difference is less than 2 cm.

**Table 4: Comparison of wave elevation amplitudes between MARS3D one-way (normal font) and MARS3D/AGRIF two-way (bold font) for the Iroise zoom (500m horizontal resolution). The amplitudes are given in cm with the relative difference in %.**





|  | **M2** | **S2** | **O1** | **K1** |
|---|---|---|---|---|
| **Point 1** | 200 (3%) | 73 (2%) | 7.0 (9%) | 7.5 (9%) |
| 5.27°W - 48.7°N | **198 (4%)** | **74 (1%)** | **6.0 (2%)** | **8.8 (29%)** |
| **Point 2** | 169 (1%) | 63 (1%) | 7.0 (6%) | 7.0 (5%) |
| 5.27°W - 48.48°N | **170 (1%)** | **63 (1%)** | **6.4 (1%)** | **8.4 (27%)** |
| **Point 3** | 179 (2%) | 66 (2%) | 7.0 (9%) | 7.0 (13%) |
| 5.5°W - 48.3°N | **180 (2%)** | **66 (1%)** | **6.5 (2%)** | **8.5 (36%)** |
| **Point 4** | 192 (2%) | 72 (1%) | 7.0 (5%) | 7.3 (5%) |
| 4.7°W - 48.3°N | **192 (2%)** | **72 (1%)** | **6.0 (6%)** | **8.4 (21%)** |

This comparison has been also done for the barotropic currents. The differences between both models are higher for the minor waves O1 and K1 with difference of 5 cm.s$^{-1}$ and relative difference of 30% for some points. However, the validation of the PREVIMER atlas currents was not available all over the area due to a lack of long time series data over the French coast. In the same idea, barotropic currents of both configurations have been compared to different available datasets. The recent ADCP data have been recorded in the Bay of Brest or in Molène archipelago. As they are mainly situated over shallow waters, there have been no significant difference between nesting techniques.

### 3.3 Improvements due to two-way nesting

A detailed qualification of the regional Bay of Biscay configuration has been done (Bezaud and Pineau-Guillou, 2015). It has highlighted the enhancements of predictions with increasing resolution in the coastal areas where the 500m zoom models have been implemented. These comparisons have been made against coarser models and the conclusions may therefore seem obvious. Indeed, a finer resolution allows the model to simulate accurately the small-scale structures (instabilities of the front, eddies, filaments...). Here, an evaluation of the two-way nesting versus one-way nesting at the same horizontal scale is performed. The two-way nesting implies the update of the mother grid fields from the child grid fields while in the one-way nesting approach there is no feedback from the child to the mother grid. The coastal Bay of Brest one-way configuration has already been validated in detail in previous studies (Frère et al., 2017; Petton et al., 2020). The two-way nesting has the same ability to reproduce the main characteristics of currents and water elevation with a high level of realism, so the comparisons are not shown. However, some differences are observed in the temperature and salinity predictions.

### 3.3.1. Hydrologic validation

For both regional and coastal configurations, the qualification of the results has been assessed over a two-year period from 2018 to 2019. Regarding the regional Bay of Biscay configuration, the evaluation focuses on the Iroise Sea zoom. Figure 6 displays a Taylor diagram for temperature (Fig. 6a) or salinity (Fig. 6b) and another diagram that represents bias values and



430    root mean squared errors (RMSE). These graphs summarize the comparisons between the available datasets and both nesting methods (one-way vs two-way) for the two different configurations. It appears that the nesting impact is not homogenous over the whole domain.

First, for the regional Bay of Biscay configuration, the two-way nesting technique improves the overall performance of the model compared to oceanic datasets (see green points for one-way and yellow points for two-way). The simulated temperature offsets are noticeably reduced. The main favourable impact is the drop of the root mean square error by 0.4°C for the COAST-HF MAREL Iroise point. For salinity comparisons, the RMSE and the bias exhibit to be in the same order of magnitude. The major improvement relies on the enhancement of correlation for the COAST-HF ASTAN dataset. This could be due to the vicinity of this point to the eastern border of the zoom. The update ability of AGRIF two-way enables more realistic incoming fluxes at high temporal resolution. The relative standard deviation of the simulated salinity at MAREL Iroise buoy is also considerably reduced. As this point is located in the strait of Brest, the tidal variations of salinity front are somehow better simulated for the same reason.

As previously stated, the one-way nesting solution is already in good agreements with the dataset (see blue points for one-way and red points for two-way) for the coastal Bay of Brest configuration. This is explained by the well-suited parametrisation performed with the bottom drag coefficient spatialisation and the vertical turbulence closure. It is also due to the fact that the flows are driven by the tide whose large-scale features are easily captured (Petton et al., 2020) in this semi-enclosed area. The two-way nesting technique barely enhances the performance of the model. For temperature, the main improvement is the reduction of bias. This could be explained by the nesting feedback that enables a more accurate global temperature budget in the mother grid. For salinity, the model also reduces the relative standard deviation for the MAREL Iroise and ECOSCOPA datasets. As the water runoffs are identical, this amelioration might be hard to explain. It may result in a better simulation of current flows with nonlinear effects.

### 3.3.2. Focus on particular processes

In the Iroise Sea during the summer, the Ushant front is depicted by cold water of about 14°C. Over shallow depths, the tidal currents are intensified and very strong around Ushant and Molène archipelago. The induced tidal stirring is so large that waters are mixed (and homogeneous) from the sea surface to the bottom. Further offshore, the summer stratification can develop and the sea surface is clearly warmer (above 18°C). This phenomenon can be seen on satellite data on Fig. 7 for the 15 of August 2016 for both Landsat 8 (Fig. 7a) and ODYSSEA (Fig. 7b) products. Compare to ODYSSEA, the Landsat surface temperature is overestimated in different spots near the coast, in the bay of Brest for example with values over 20°C. This might be due to mis-flagged clouds.

The ability of the two-way nesting approach (Fig. 7d) to correctly reproduce this spatial feature is clear while the one-way nesting (Fig. 7c) struggles to simulate this phenomenon. Indeed, the Ushant front is nearly missing in the one-way simulation and is much better characterized in the two-way simulation at 5.25°W. The temperature magnitudes are strengthened on each side of the front thanks to the AGRIF update. Furthermore, this improvement supports the realism of the fine structures around





the shoals of Sein Island and in the Molène archipelago in the two-way simulation. On another hand, one could notice the open boundary effect in the south part of the one-way simulation where an east-west temperature front is created.

## 465    **3.4. Flushing times of the Bay of Brest**

The question addressed here is to identify the role played by the tidal forcing and water runoffs on the renewal capacity of the bay under consideration. This renewing process takes place over several days. Although wind direction and intensity are highly variables at mid-latitude, it has been decided not to focus on meteorological effects because it is difficult to find 15-days wind sequences that characterize the local atmospheric forcings. In addition, the bay is a highly energetic coastal area with strong
diffusive and dispersive characteristics as its regime is macro-tidal.

Instead, the focus is on the two dominant runoff regimes (flood water *vs* low water) combined with the initial phase of the tide (spring *vs* neap tides). The flushing times of the whole volume have been estimated for each scenario and are given in Table 5. For each simulation, this timescale features the lowest value for a release at low tide with a maximum variation of 1.4 day. As expected, low water runoffs imply larger renewal time compared to flood situation. However, the initial tidal phase is the
main factor of change between the scenarios with a more intensive mixing in spring tides. There is a positive offset (roughly 10%) when the AGRIF two-way nesting is used for the estimation of this indicator. This is due to the return flow inside the bay during each tidal cycle that is under-estimated in the one-way model as its boundaries act as a sink for the tracer.

**Table 5: Global *e*-flushing times and standard deviation in days for the whole control volume for both modelized configurations. The deviation is estimated over the 13 time released simulations for each scenario.**

| Scenario | Global *e*-flushing time in days | |
|---|---|---|
| | one-way | two-way |
| Low water – Neap tide | 11.29 ± 0.42 | 12.23 ± 0.64 |
| Low water – Spring tide | 7.68 ± 0.39 | 8.34 ± 0.48 |
| Flood – Neap tide | 10.22 ± 0.41 | 10.84 ± 0.53 |
| Flood – Spring tide | 6.95 ± 0.35 | 7.85 ± 0.53 |


The local *e*-flushing times estimated with the two-way nesting configuration in low-water conditions are displayed for initial spring tide (Fig. 8a) and neap tide (Fig. 8b). In the same way, the renewal indicator is shown for flood conditions for initial spring tide (Fig. 8c) and neap tide (Fig. 8d). For the four scenarios, the coefficient increases with the geographic position relative to the border of the control volume. In opposition to the computation of global indicator, here the impact of runoff is
clearly visible in the renewal capacity of the bay. In low water condition, the south-eastern part of the bay is clearly isolated from the rest. The range of local *e*-flushing time reaches more than 25 days in shallow coves for a release at neap tide. The





impact of tide is the next level in order of importance with, not surprisingly, stronger renewal during spring tide than neap tide. In each scenario, the central energetic eddy stands out because it is the rallying point of continental waters.

Some particular structures appear at the western part of the control volume. This is due to a weak exponential regression where

the residuals between exponential decay and simulation result are below 0.5. This part of the domain was only there to account for the tidal prism extent. A similar issue occurs next to the river mouths. Both problems are due to the initial time release over the first tide cycle. This is easily understandable for areas next to the volume limits in the ocean. For river mouth, de Brauwere et al. (2011) pointed out than initial time release in estuaries could lead to extreme different values, as it is here between a release at high tide and low tide. However, the fact of considering initially the flushing lag in each mesh strengthened

considerably the *e*-folding regression approach.

The same analysis has also been performed with the traditional one-way nesting configuration for the four scenarios. For each mesh, the relative differences are shown in Fig. 9 in the same order in Fig. 8. First, spring (Fig. 9a) and neap (Fig. 9b) tides in low water condition and second spring (Fig. 9c) and neap (Fig. 9d) tides in flood condition. The one-way nesting overestimates in each cases the renewal capacity of the bay by predicting lower local *e*-flushing time nearly everywhere. In the inner part of

the bay, the local differences are always negative and reach around 20% in flood conditions. Under low water condition, although there are no changes for the energetic water mass in the central part, the discrepancies appear in the eastern part of the bay and in shallow areas. Here again for each scenario, the structures at the limits of the control volume are not clear for any scenario and present sharp extents with alternation of under and over-estimations.

## 4. Discussion

The present study demonstrates the capabilities of MARS3D/AGRIF two-way nesting compared to classical one-way offline embedding; it is based on gradually more complex examples from local to global issues. The very first objective of grid refinement is either to tackle a local stationary problem or to follow a single dynamical structure along to its displacements (Blayo and Debreu, 1999). The zoom is there to reach a relevant resolution (*e.g.,* commensurable with the Rossby internal radius or with the geometry of a given structure or set of structures such as islands, capes or peninsulas) raising the major

question which lies in the downscaling of global and coarse information to the resolution of the targeted area at least at the open boundaries of the considered domain.

Despite the regular improvement in the computational power available for high-performance scientific computing, the computational cost and the storage of huge datasets remain major issues for long-term numerical oceanic simulations, on behalf of various reasons such as green computing considerations. We hereafter review the different key advantages provided by the

AGRIF library as used for a split free surface ocean model set on a structured grid, which is a fairly common tool in the coastal oceanic community.

First, the usual offline nesting procedure requires to write and store the 3D forcing file from which the open boundary and initial conditions for the child grid are picked up. In order to avoid aliasing or spurious incompatibilities between barotropic



and baroclinic modes, high frequency writings to hard disk are required; in the most penalizing cases, for example where tidal
dynamics are dominant, it involves a huge amount of I/O, which raises other kind of issues such as how to write massive data
in a massive parallel context. Despite there are many improvements to deal with that question like deporting the I/O on
dedicated CPUs as it is the case in the XIOS library (Yepes-Arbós et al., 2022), they cannot escape the cost of long-term
storage of massive data. This is even more obvious in case of several nested grids in the same mother grid (such as the Bay of
Biscay configuration) which would extend the pre-processing step to create child boundaries and require more flexibility which
sketched what AGRIF may provide. The on-the-fly grid nesting procedure (encompassing initial and open boundaries
management) included in the two-way nesting circumvents these tedious steps by performing them online at each time step,
and for the different grids of the hierarchy.

The second point concerns the vertical coordinate framework which is here a sigma framework. The vertical interpolation
towards a sigma framework requires most of the time the projection onto a geopotential framework in order to perform the
split horizontal-vertical interpolation: the horizontal interpolation consists in the rescaling from the coarse to the fine horizontal
grid whereas, the vertical interpolation mainly consists in a resampling of the vertical levels without any changes in the overall
resolution (or the use of a kriging method). The side effects of the split interpolation on temperature and salinity fields can
lead to gravity instabilities in case of large bathymetric inconsistencies between the coarse and child grids. Therefore, the user
hast to carefully define interpolation parameters (such as those defining the intermediate geopotential framework onto which
the vertical interpolation is performed) and check the consistency of the gravity gradient. As MARS3D/AGRIF two-way
nesting requires a perfect fitting of the vertical discretization of all grids of the hierarchy and bathymetries coherence, these
constraints prevent from gravity issues. This kind of very well-known problem may also be avoided in offline embedment
procedure by taking the same cares in the grid and the bathymetry computations. Nevertheless, the heavy constraints imposed
on the vertical coherence of the grids induces a lack of flexibility that may leads to absurdly over-resolved vertical grids; this
is particularly the case when the bathymetry of the parent grid extents from the coast to the abyss, requiring much larger
number of vertical levels in the abyss than would have been necessary for a standalone child grid restricted to the continental
shelf. Reducing the number of vertical levels from mother to child grid may save precious computing power when resolving
vertical dynamic process in well-mixed areas.

As shown in Sect. 3.2, the online interpolation from mother to child grid and the update process preserve the propagation of
tidal elevations and currents with an equivalent level of accuracy than the one reached with the tidal forcing prescribed at open
boundary conditions of a single typical coastal grid (some tens of kilometres of extension, in 10 to 100 m depth, at some
hundreds of meters of resolution). For that kind of standalone grids, validated harmonic atlas at high horizontal resolution (*i.e.*
from 750 m to 250 m, Pineau-Guillou, 2013) may be available: they enables to represent accurately the tidal elevations and
currents within the encompassed area thanks to adapted open boundary conditions algorithm (Flather or characteristics).
According to the slow varying characteristics of the tidal components this downscaling approach can be thought to be
performed once for good by generating a reference tidal atlas. In the two-way nesting approach depicted here, the tidal
propagation computed at the coarse grid level is fed with a global reference tidal solution, either TPXO (Egbert and Erofeeva,





2002) or FES (Lyard et al., 2021). For the regional configuration exposed previously, the tide coming from the mother grid with the two-way nesting has been only computed with the global FES model. The observed differences between the one-way

offline and the two-way nesting methods are less than a few centimetres. They are not straightforward over the coastal area and even not really significant. It relies mainly on the continuity of the mass fluxes at the interface between the child and the coarse grids. This functionality is preserved thanks to the AGRIF library implementation and maintains MARS3D good ability to represent tides at regional scales and medium resolution (Lazure et al., 2009; Muller et al., 2014).

For both configurations exposed previously, the results shown in Sect. 3.3 highlight the enhancements of the two-way nesting

hydrodynamic simulations compared to the one-way ones. First, for wide open areas such as Iroise Sea, the benefit of the feedback/update of the zoom contributes to the correct representation of the thermic Ushant front mainly by enlarging the span of scale captured (*e.g.,* fine eddies and filaments), which is expected, but also in term of the horizontal localisation of thermal front, which is less expected. As a matter of fact, the forcing through the open boundaries is of primary importance for limited area models; it may dramatically impact the local coastal processes, even in areas where the dynamics is highly controlled by

large scales as the tidal forcing. The contribution of the two-way method to the improvement of the offline one-way nesting is less obvious for the case of Brest, a semi-enclosed bay. The two-way nesting corrects a global bias of 0.15°C for temperature which could be linked to the high frequency hydrologic forcing at boundaries. Otherwise, the two-way method only slightly improves comparison statistics on temperature or salinity within the bay, despite the fact it is subjected to large exchanges with the open sea; its relatively shallow depth along with highly variable turbulent mixing make the dynamics less sensitive to

boundary effect.

On the contrary, when quantities along the boundaries are less homogeneous or when their time series is rather complicated (not straightforwardly predictable with the tidal signal), the method is more valuable. It is illustrated in the Sect. 3.4 with the computation of renewal time indicators on the same configuration of the bay of Brest. The improvement of the AGRIF two-way nesting consists both in the refining of coastal areas where water renewal is the most critical and in properly tracking the

tidal prism enclosed in the global, advected and time-dispersed control volume. The known methods used to estimate such parameters are generally constraint to the definition of a global control of volume. This volume is tightly related to the geographic context and the environmental study objectives. By taking into account the flushing lag and estimate the *e*-local flushing time with an exponential regression, the final indicator is more local and less sensitive to the initial released volume. Nevertheless, the management of the open boundaries for the fine grid is a key element to address properly this issue. In the

case of a standalone grid, when the flow is entering the domain at the open boundaries, the user can usually either chose to apply or nudge towards a constant value (zero or whatever value), a time series inferred from large scale solution (if available) or a zero-gradient condition. In any case, the choosen solution is highly non conservative, resulting in discrepancies for the final flushing indicators. The two-way nesting enables the correct estimation of local flushing parameters in the Bay of Brest, which requires accurate tracking of the tidal prism encompassed in the control volume across time. Here, a significant part of

water is tidally flushed of the bay and spread to the Iroise sea with diffusion processes. Concepts of flushing time need to consider the back-and-forth flow into the control volume over long time scales (*i.e.,* scales that largely exceeds the scale of the



main driving process) thus requires a larger simulated domain compared to the control volume as stated by Viero and Defina (2016a). The AGRIF capability improves the management of the fate of a tracer outside the bay at the coarser horizontal resolution (and at fairly reasonable computational cost) and enforces accurate incoming concentration along the open
boundaries on the child grid. This conservative approach required for an accurate estimation of the indicators is also relevant for applications with sedimental, biological or chemical dynamics models that can be coupled.

Regarding parametrization, the AGRIF library flexibility allows to specify a distinctive parametrisation for each grid of the hierarchy (meteorological forcings, runoffs, vertical turbulence scheme, surface and bottom friction coefficients…). In addition of the different physical parametrizations well suited to local process, a local time refinement is performed to take into the
local dynamics: each grid of the hierarchy can have its own sub-temporal integration as used in the regional seven-zooms regional configuration (ratio of 3 to 5). The more the dynamics is intense the more the time refinement factor can be increased. Whereas for child areas where the stability condition is weaker, the time refinement factor can be diminished.

The AGRIF flexibility also lies in the way child grid in a given hierarchy can be either added or easily removed: the addition of a grid requires the coupling of this additional child grid bathymetry with the bathymetries of the other grids. An offline
bathymetric update algorithm, available with MARS3D code, modifies both mother and child bathymetries in order to fit fluxes (from one grid to another) along child borders. Once the bathymetry consistency is performed along the boundaries between the mother and child grids, the user can launch the runs without additional task. The initial conditions for the new zoom are estimated online at the first timestep by the AGRIF library. This capability has been used to study more precisely the deep convection in the northwestern mediterranean sea (Garreau and Garnier, 2015) or to better identify shear stress over
future renewable marine energy structures (Maisondieu, Pers. comm). In situation where the user needs to remove a specific child grid, the bathymetry consistency has just to be checked once offline before launching the model.

This software also has the capability to perform the opposite, that is to say grid coarsening. It is a relevant capability for on-line physical-biogeochemical coupling which was used for example by Lévy et al. (2012) in an offline mode. Capturing relevant scales of the oceanic flows may require higher spatial resolution than the main feature of biogeochemical fields. In
such perspective, it is essential to solve the submesoscale features that enhance the vertical exchange and consequently refuel the surface with nutrients. The required resolution, of the order of a few hundred meters, is unaffordable for state-of-the-art biogeochemical models that use and advect tens of tracers fields (Heinze and Gehlen, 2013). The AGRIF coarsening capability allows this differential resolution (physics at high-resolution, biogeochemistry at lower resolution) by building on-line non-divergent transport fields from the high-resolution grid on the coarse resolution grid. In that way, the "grand-mother" grid may
completely overlap the mother grid.

In contrast, another perspective for the AGRIF library is to generalize the refinement approach towards what is called a multi-resolution approach (Debreu et al., 2016): adding not only a few grids but tens to hundreds of grids in order to pave the space systematically according to a general refinement criteria. Even if the ocean is turbulent everywhere, it is known that the computational grid should be refined systematically next to the coast for multiple reasons: the Rossby radius is thinner, there
are physical obstacles such as capes, islands, isthmuses, tributary, reefs, or there is a quite great continuity in the fate of coastal



discharges that nevertheless follow tortuous paths along the coast. One solution consists in using unstructured grid which are steadily refined from offshore to the coast. This alternative involves the creation of a thin pavement next to the coast which is encompassed in a coarser pavement, with a refinement limited to a factor 2 to 5. A simple example built on our area of study is exhibited in Fig. 10 with a 500m horizontal resolution for the finest grids. This pavement is set with a hierarchy of three
levels and with a space refinement factor of two (one mother grid in blue, 19 child grids at the first level in green, 61 child grids at the second level in red); the used criteria combine the distance to the coast and bathymetric thresholds. The integration of numerous grids of various size with various overlapping is not straightforward in massive parallel computational framework and raises many questions of load balance, processors dispatching, communication blocking, etc… Another concern is the data structure and the format used for the I/O in order to perform easy writing and to perform postprocessing diagnostic and
visualisation.

## 5. Conclusions

Algorithms for the implementation of two-way structured mesh refinement in an implicit free surface ocean model are described in this paper. For coastal applications, the correct treatment of wetting and drying requires a careful design of the grid interactions. Applications of such a mesh refinement technique allows to reach very high-resolution all along the French
coast of the Bay of Biscay featured by a macro-tidal regime. The present analyses demonstrate the positive impact of the two-way refinement against simple one-way refinement. In addition to the applications presented here, these mesh refinement capabilities have now been validated on a series of realistic configurations.

The computational aspects of grid refinement are managed by the AGRIF software. In the context of sustainable green computing, this nesting technique saves tedious file processing and interpolation steps. It is well suited for large configurations
with several zoom models, for long-term hindcast or operational forecast simulations to monitor marine environment. The AGRIF implementation in the MARS3D model give the opportunity to the users to easily activate or deactivate a given child grid. Adding one additional zoom level is a quick process which actually relies more on the accuracy of the available bathymetric data than on technical issues. In coastal areas, the two-way nesting solution enables the correct estimation of environmental indicators calculated for highly different hydrologic and tidal forcing in order to characterize the general mixing
of the bay. It allows the fine simulation of biological processes that require mass conservation at large scales with different horizontal resolution.

## Code availability

Last version of MARS3D is freely available on request at https://wwz.ifremer.fr/mars3d/. The AGRIF library is freely available at under CECILL license (http://www.cecill.info). Both codes are written in Fortran-90/95 and figures are displayed from
python scripts or with QGIS software for both configuration presentations. All the model code, bathymetric grid files, namelist



configuration files for both regional and coastal applications and python scripts used in this paper are available at
https://doi.org/10.5281/zenodo.6672562.

**Data availability**

All data used in this paper are freely available from their DOI repository.

**Author contribution**

LD has developed the AGRIF Library. LD, VG and FD have integrated the AGRIF library in the MARS3D model. MC and
SP have setup the model configurations, adapted the AGRIF integration to coastal environment and provided figures for the
paper. All authors have contributed to the concepts and the writing of the paper.

**Competing interest**

The contact author has declared that neither he nor their co-authors have any competing interests.

**Financial support**

V. Garnier, F. Dumas and L. Debreu were funded by the French National Research Agency (ANR) through (COMODO).
MARS3D - AGRIF capability has been implemented and funded within the framework of PREVIMER project to monitor the
regional and coastal environmental dynamics.

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





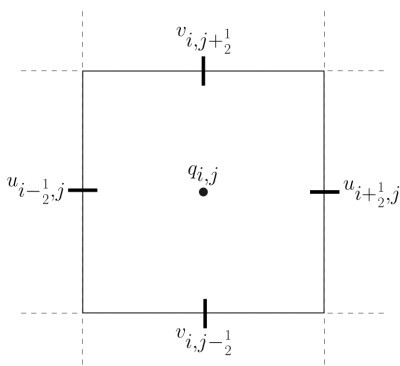

**Figure 1: Staggered Arakawa C grid. $q$ stands for pressure and tracer points while $u$, $v$ are the velocity components.**

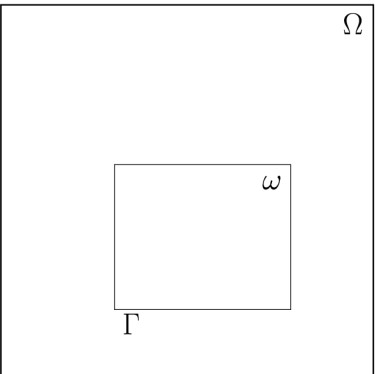

**Figure 2: Local refinement**

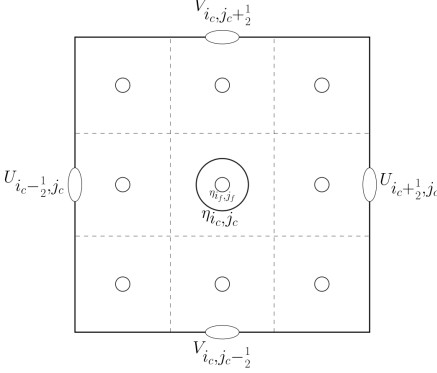

**Figure 3: A coarse grid cell divided in nine fine grid cells on a C grid**





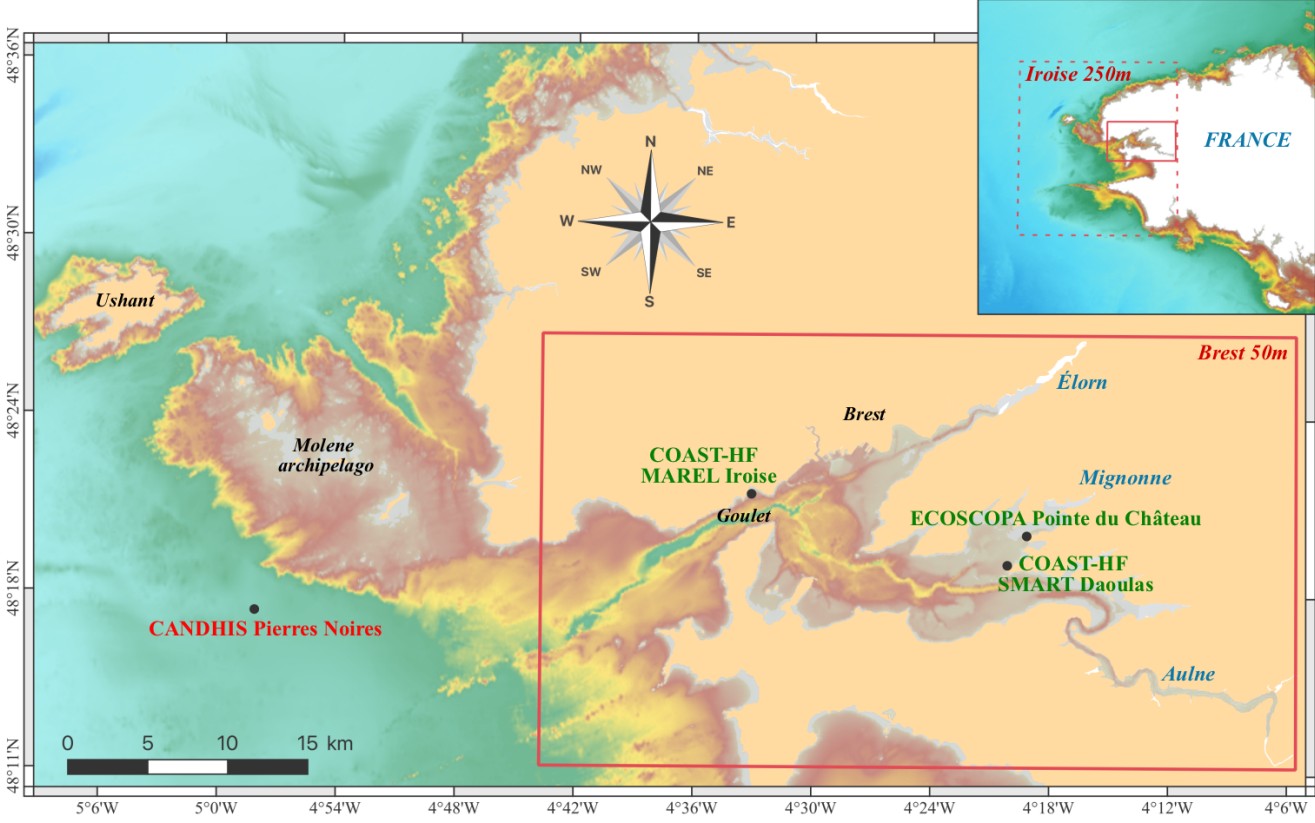

**Figure 4: Bay of Brest configuration. The geographic extent of the zoom grid at 50 m resolution is the solid red rectangle. The coarser grid at 250 m resolution is the dashed red rectangle. Bathymetric and coastline sources: Ifremer / SHOM.**





**Figure 5: Bay of Biscay configuration with seven zooms of 500m resolution (red rectangles). The 2.5 km resolution coarser grid (orange rectangle) is included in a larger 2D model at 5 km resolution (blue rectangle). The sea surface temperature is given for the 16th august 2018 with the finest possible resolution. Bathymetric and coastline sources: Ifremer / SHOM / Natural Earth.**



**Figure 6: Temperature (a) and Salinity (b) validation for both configurations. The Taylor diagrams are represented with relative**
**standard deviation (blue dashed lines), correlation (green dashed lines) and relative root mean square error (red dashed lines).**



**Figure 7: Sea surface temperature over the Iroise sea on August 15th, 2016 for Landsat 8 (a), Odyssea (b), one-way simulation (c) and two-way simulation (d). Coastline source: SHOM.**




**Figure 8: Local *e*-folding flushing time estimated for spring (a) and neap (b) tides in low-flow conditions and for spring (c) and neap (d) for flood conditions. Coastline source: SHOM.**






**Figure 9: Differences between local *e*-folding flushing times estimated with one-way configuration over two-way nesting method. They are computed for spring (a) and neap (b) tides in low-flow conditions and for spring (c) and neap (d) for flood conditions. Coastline source: SHOM.**






**Figure 10: Multi-resolution approach with three levels of refinements and a space refinement factor of two. The mother grid (2 km horizontal resolution) limits are in blue, 19 child grids at the first level (1 km horizontal resolution) in green and 61 child grids at the second level in red (500 m horizontal resolution). The criteria used for clustering combines the distance to the coast and bathymetric thresholds. Bathymetric and coastline sources: Ifremer / SHOM.**
