# Peer review of "Using two-way nesting technique AGRIF with MARS3D V11.2 to improve hydrodynamics and estimate environmental indicators"

_EGUsphere, 2022_

## Referee Comment (RC1)

Review comments on the preprint titled:

**Using two-way nesting technique AGRIF with MARS3D V11.2 to improve hydrodynamics and estimate environmental indicators**

by Sébastien Petto, Valérie Garnie, Matthieu Caillaud, Laurent Debreu and Franck Dumas

The article describes an application of the integrated MARS3D-AGRIF model for two-way nested coastal modelling, particularly for estimation of environmental indicators.

**Major points:**

1. The article looks like a project report as technical details and data information take quite a lot of space. It is a bit short of novelty but an extended application of existing model techniques. As the authors have stated that the AGRIF software has already been implemented into the MARS3D model and two-way nesting has been reported by Debreu et al (2012). Authors may try to restructure the article and clarify what has been done before and what is done in this study, minimising duplication of published materials.

2. Whether this technique is suitable for coastal marine application is still a question though the authors have claimed it is 'well suited'. There are three reasons for this questioning:

a) The two-way interaction is only built between the mother grid and each child grid. Although it could create a chain of child grids along coastlines and some child grids may even overlap, there is no direct interaction among the child grids, neither two-way nor one-way. This casts a doubt on whether the child grids could really represent the coastal processes as whole. The mixed effects reported by the article may be evidence.

b) AGRIF is designed for adaptive refinement in open ocean, may not be a good approach for coastal refinement. The final example of 3-level refinement with 1 mother grid at 2 km, 19 grids at 1 km and 61 grids at 500 m resolutions is complicated as a coastal model. Nevertheless, the 3-level refinement may not be enough for coastal models. For instance, the 500 m fine resolution may not be enough to resolve the 1.8 km wide 50 m deep narrow channel (Goulet de Brest). The 2 km mother grid may not be suitable for a large regional model to cover key regions desired for far-reaching effects. It looks the software may struggle if more levels are required.

c) Although the AGRIF software has removed the burden to save boundary files for the child grids, there is still a requirement to save the output from these child grids. Otherwise, the enhancement by the refinement will be lost. The duplication of data over same areas (as the mother and child grids overlap) also increases disk demand, not to mention that some child grids may also overlap, and the rectangular domain shape wastes some disk spaces for land areas.

3. Parallelization is an important feature of large models, like this coastal application. The article does include some description of parallelization but may extend it to give more details, at least, to clarify some results as given in section 3.1.

**Minor points**:

L40-51: This paragraph stated that it is difficult for unstructured grids to handle adaptive refinement both temporally and spatially but failed to clarify that the present application is not adaptive as well though the AGRIF software is originally designed for adaptive mesh refinement. There is also multi-resolution unstructured grid, such as the spherical multiple-cell grid (Li 2021),

which can offer static coastal refinement and prevent the finest cell CFL restriction from spilling over the entire grid.

L53: It is a bold claim that overlapping grids would be "well suited" for coastal applications.

L75: The section title is ambiguous. Using the sub-section titles directly may be good enough.

Section 2.1: An important feature of a 3-D marine dynamical model is the vertical discretisation or vertical coordinate scheme. The sigma coordinate used in the MARS3D model is better to be mentioned in this section rather than quite later in section 3.1. Authors may simplify the MARS3D model description, such as shortening the description of the C-grid as it is well documented, and concentrate on any update for this study, such as any modified equations or changed schemes.

Section 2.2: Similar as comment for section 2.1. Is the two-way nesting algorithm differ from Debreu et al (2012) one? If so, what is new in this model implementation?

Sub-section 2.2.3: Parallelization is an important feature of large models. Better start a new section for it and explain a bit more about it.

L195: Do you mean distributing the MPI processors among all child grids?

Sub-section 2.2.4: Is this part of the two-way nesting algorithm? If so, better merge it into sub-section 2.2.1.

Section 2.3 and 2.4: These sections could be reduced to essential information, so they are just enough for demonstrating the two-way nesting effect.

L227: Move the 3 in 'm3' to a superscript.

Section 3.1: May change the section title to: Computing cost. It may serve as part of the model parallelization. Explain more on how many runs are used for the computing time average and clarify the computing node usage, particularly for the hybrid one.

L368: "computing coast" should be computing cost".

Table 2: what do you mean 56 MPI nodes with 8 OMP each? Do you mean the total number of nodes are 56x8? Better to clarify in terms of MPI ranks and OMP threads as number of processors on one computer node may vary from machine to machine. Reducing the runtime to 1/4 with 8 times of computing resources in hybrid mode is a good result.

Section 3.2-3.4: Limited by my knowledge about coastal models and environment indicators, I could not comment much on these sections.

Section 4. Would it be better to merge the discussion with the result section?

L616-630: The paragraph for possible multi-resolution approach with AGRIF library may be shortened as it is not part of this study. Fig.10 gives readers a false impression that the 3-level configuration has been tested. It is better removed.

L700: Better not quote conference abstract with a broken web link.

**Reference**

Li, J.G. 2021: Filling oceans on a spherical multiple-cell grid. *Ocean Modelling*, **157**, 101729. DOI: 10.1016/j.ocemod.2020.101729

---

## Author Comment (AC1)

**Response to reviewer 1**

We first want to thank you for paying very careful attention and spending time to review our work; we hope these modifications make the manuscript clearer and improved it sufficiently to render it worth publishing.

The article describes an application of the integrated MARS3D-AGRIF model for two-way nested coastal modelling, particularly for estimation of environmental indicators.

*Major points:*

1. The article looks like a project report as technical details and data information take quite a lot of space. It is a bit short of novelty but an extended application of existing model techniques. As the authors have stated that the AGRIF software has already been implemented into the MARS3D model and two-way nesting has been reported by Debreu et al (2012). Authors may try to restructure the article and clarify what has been done before and what is done in this study, minimising duplication of published materials.

We synthetize the technical details to avoid the "project report" aspect of what was written in the first submission and develop the numerous novelties that have been implemented in the systems since the cited papers. Despite the fact that AGRIF has been developed for years (almost decades) and has been implemented in a couple of models such as ROMS-AGRIF (Debreu et al., 2012), NEMO (Biastoch et al., 2018), MARS3D (Dufois et al., 2014), the model MARS3D V11.2 with AGRIF and the applications performed illustrated new capabilities and open rooms for future conceptual developments. It may be first worth to stress the fact that the approach relies on the development of a code-independent library (AGRIF), suited for structured grids and finite differences/ finite volumes methods, and weakly intrusive into codes (Debreu and Blayo, 2008). In that way, it offers a generic capability implementable in any code fulfilling the prerequisites nevertheless requiring a good knowledge of code numerical aspect (time stepping, advection scheme, mode splitting).

As a matter of fact, the coupling reported by Debreu et al. (2012) relies on the ROMS-AGRIF model. In many ways ROMS-AGRIF differs from MARS3D: mostly it is a split explicit free surface model (Shchepetkin and McWilliams, 2005) whereas MARS3D is a split semi-implicit free surface model (Lazure and Dumas, 2008). The use of the Alternate Direction Implicit (ADI) which is a global solver, solving a full row or column of the grid in one step induces many tricks especially in the two-way implementation that have been addressed since the first application of Muller et al. (2009). Theses aspects has been developed in the reviewed manuscript.

Here we can mention specifically what has been added or better highlighted:
- The reformulation of the ADI scheme to solve the barotropic mode,
- The update method between a given child grid and its parent grid, that insure momentum and mass conservation,
- The communication between overlapping child grid of the same level of the hierarchy,
- Some specific point related with the wetting and drying capability.

As you noted, an older version of the AGRIF library was already implemented in MARS3D V6.0 and used in Muller et al. (2009) and Dufois et al. (2014). This implementation solely assured the mass conservation with one-way nesting. The MARS3D-AGRIF V11.2 code gathers numerous developments. We understand it could be a bit confusing as it represents nearly a continuous 5-years period of development. Considering this remark and your next suggestions, the article was restructured to better identify what is new in MARS3D model, AGRIF library and the two-ways nesting technique implementation. The Sect. 2 is now devoted to innovative developments with reference to previously published studies.

2. Whether this technique is suitable for coastal marine application is still a question though the authors have claimed it is 'well suited'. There are three reasons for this questioning:

a) The two-way interaction is only built between the mother grid and each child grid. Although it could create a chain of child grids along coastlines and some child grids may even overlap, there is no direct interaction among the child grids, neither two-way nor one-way. This casts a doubt on whether the child grids could really represent the coastal processes as whole. The mixed effects reported by the article may be evidence.

The requirement pointed here is very relevant and we paid much attention to it despite the fact it was apparently not sufficiently described in the first version of the manuscript. It led to a misunderstanding and thus we clarify the manuscript. The child grids are indeed enabled to exchange information thanks to the overlapping areas as it was explained in paragraph L208-215 (subsection 2.2.4). We extended the description of this interaction between grids of the same level through their common parent grid. Without these interactions at the same hierarchical level, choosing to pave the coastal domain with rectangular shapes such as in the regional configuration would have been an inadequate solution. The overlapping areas are necessary to apply a weighted-average operator between child grids.

b) AGRIF is designed for adaptive refinement in open ocean, may not be a good approach for coastal refinement. The final example of 3-level refinement with 1 mother grid at 2 km, 19 grids at 1 km and 61 grids at 500 m resolutions is complicated as a coastal model. Nevertheless, the 3-level refinement may not be enough for coastal models. For instance, the 500 m fine resolution may not

be enough to resolve the 1.8 km wide 50 m deep narrow channel (Goulet de Brest). The 2 km mother grid may not be suitable for a large regional model to cover key regions desired for far-reaching effects. It looks the software may struggle if more levels are required.

You are totally right that the example given is not refined enough in coastal areas. Resolution below hundred meters is now an order of magnitude to study coastal physical or biological processes. Actually, the pavement was described as "a simple example" to illustrate the capability of geographic domain splitting. Nevertheless, the MARS3D-AGRIF system is able to cope with more complex grid hierarchy: We provide here an example with two levels of grid refinement but there is formally no limit on the number of levels in the grids hierarchy. We could have built a hierarchy with three, four even more levels to downscale up to any given resolution. So far, the main problem raised by such an approach relies more in the postprocessing (of numerous file results) than in the building nor the computation. This problem is being addressed for a next generation of our approach.

On another hand for biological studies where more than 20 tracers are simulated (Ménesguen et al., 2018), this example configuration (already tested for a short period) still represents a suitable opportunity to pave the French coast at reduced computation cost without jeopardizing the final objectives.

Because this development is still an on-going perspective, we decide to reduce it with just a short mention.

c) Although the AGRIF software has removed the burden to save boundary files for the child grids, there is still a requirement to save the output from these child grids. Otherwise, the enhancement by the refinement will be lost. The duplication of data over same areas (as the mother and child grids overlap) also increases disk demand, not to mention that some child grids may also overlap, and the rectangular domain shape wastes some disk spaces for land areas.

We completely agree: a new structured output data needs to be elaborated, to avoid duplicated writing data at different refinement scales over the same geographic areas. An INRIA team is developing such output design bearing in mind the compatibility with processing software (Python, Paraview…) to easily interact with. It is based on different merges of grids at same level without keeping data where there is a subgrid level. Another solution would be to use the same as unstructured model's output file with 1D list containing geographic position, hierarchical grid level and spatial relations with adjacent cells. This issue was already mentioned in L628-630 so we emphasize it by explicitly telling that further improvements are still required.

Concerning the disk space used for land areas, the rectangular domain shape waste remains minimal as all land values are declared as a missing value thanks to the _Fill_Value attribute of the NetCDF norm. With the classic on-the-fly compression using zlib (https://zlib.net) integrated into the NetCDF4 library, the output file's final size is largely reduced, almost as if these points do not exist. One may even think to mask in a parent grid all the areas overlapped by a child grid (not yet performed in our system).

3. Parallelization is an important feature of large models, like this coastal application. The article does include some description of parallelization but may extend it to give more details, at least, to clarify some results as given in section 3.1.

We do agree; some words were given in the section 2.2.3 but certainly not sufficient regarding the importance of this aspect. We extend the description of the parallelization and we comment in further details the results in section 3.1.

***Minor points:***

L40-51: This paragraph stated that it is difficult for unstructured grids to handle adaptive refinement both temporally and spatially but failed to clarify that the present application is not adaptive as well though the AGRIF software is originally designed for adaptive mesh refinement. There is also multi-resolution unstructured grid, such as the spherical multiple-cell grid (Li 2021), which can offer static coastal refinement and prevent the finest cell CFL restriction from spilling over the entire grid.

We do agree and are thankful for the reference that help to mitigate our initial statement. Our point is mostly focused on the temporal refinement which is very easy to manage from one grid to another (one may notice it is constant on a given grid). We did not intend to address here the problem of the spatial adaptive refinement which is, as you underline, out of the scope of this study where we use static grid hierarchy. We are more in line with Li (2021) approach to better fit a fixed geometry rather than tracking a dynamic feature (such as an oceanic eddy, a river plume, a frontal instability). We also add that to our (almost comprehensive) knowledge AGRIF is not use for application with its spatially adaptive capability relevant to follow dynamical patterns of a flow.

L53: It is a bold claim that overlapping grids would be "well suited" for coastal applications.

You are right. The overlapping is compulsory to exchange information between child grids and might not be claim as "well suited". We underline this and temperate what might appear as a "bold claim".

L75: The section title is ambiguous. Using the sub-section titles directly may be good enough.

As mentioned earlier, we split Sect. 2 into two parts: the first one is dedicated to innovative model developments and the second one is focused on both realistic configurations and timescale indicator description. Thus, Sect. 2 is now named "Innovative developments for two-way nesting".

Section 2.1: An important feature of a 3-D marine dynamical model is the vertical discretisation or vertical coordinate scheme. The sigma coordinate used in the MARS3D model is better to be mentioned in this section rather than quite later in section 3.1. Authors may simplify the MARS3D model description, such as shortening the description of the C-grid as it is well documented, and concentrate on any update for this study, such as any modified equations or changed schemes.

We do agree with you. We dramatically simplify these parts with only the newest developments and remind that MARS3D is based on sigma vertical coordinate framework (itself rather common and well documented but worth to be mentioned).

Section 2.2: Similar as comment for section 2.1. Is the two-way nesting algorithm differ from Debreu et al (2012) one? If so, what is new in this model implementation?

Yes, it is significantly different as some fundamental numerical aspects of these two models differs noticeably, especially the time stepping and the baroclinic/barotropic coupling. The main one relies on implementation in split explicit free surface model (ROMS-AGRIF as in Debreu et al. (2012)) *vs* split semi-implicit surface model (never performed before to our knowledge). We think it is worth presenting it as long as the ADI (*ie* semi-implicit) solver used in MARS3D is quite popular among coastal modelling community (Chakraborty et al., 2021; De Goede, 2020; Parsapour-Moghaddam and Rennie, 2017) concerning model as widely spread as Delft3D or Mike21. As said earlier, the recent developments have now been highlighted in the Sect 2.

Sub-section 2.2.3: Parallelization is an important feature of large models. Better start a new section for it and explain a bit more about it.

According to your rightful suggestion, we create a new subsection 2.3 dedicated to the parallelization option and give more details.

L195: Do you mean distributing the MPI processors among all child grids?

Yes, it is exactly what we meant. We replace the term "between each" with "among all".

Sub-section 2.2.4: Is this part of the two-way nesting algorithm? If so, better merge it into sub-section 2.2.1.

The definition of a sponge layer applied in the child grids is indeed a part of the two-way nesting algorithm. Following your suggestion, we merge it with subsection 2.2.1. The second part of the initial subsection was dealing with the interaction between child grids at the same hierarchical level; we keep it in a special subsection to highlight this important issue (as you underlined above in your second major point).

Section 2.3 and 2.4: These sections could be reduced to essential information, so they are just enough for demonstrating the two-way nesting effect.

We decided to create a new section with these subsections. We have reduced the description of forcing and data validation as recommended and avoid the "project-report-like effect" it gives.

L227: Move the 3 in 'm3' to a superscript.

Ok.

Section 3.1: May change the section title to: Computing cost. It may serve as part of the model parallelization. Explain more on how many runs are used for the computing time average and clarify the computing node usage, particularly for the hybrid one.

We gather a part of this section with the parallelization section. We also clarify the paragraph and correct the wrong usage of term node. Five simulations were run sequentially to determine the average computing time.

L368: "computing coast" should be computing cost".

Ok

Table 2: what do you mean 56 MPI nodes with 8 OMP each? Do you mean the total number of nodes are 56x8? Better to clarify in terms of MPI ranks and OMP threads as number of processors on one computer node may vary from machine to machine. Reducing the runtime to 1/4 with 8 times of computing resources in hybrid mode is a good result.

OK. We definitely need to clarify this part as we made some confusion with the term "node" instead of using "ranks". The supercomputer DATARMOR is composed of 396 nodes with 28 cores each. When we were saying "56 MPI nodes with 8 OMP each", it should have been "56 MPI ranks with 8 OMP threads, for a total number of 448 cores". We modify Table 2 according to this terminology. Indeed, the hybrid mode is efficient to reduce the computing time. We highlight this remark in the manuscript.

Section 3.2-3.4: Limited by my knowledge about coastal models and environment indicators, I could not comment much on these sections.

Section 4. Would it be better to merge the discussion with the result section?

We sum-up the description of the results to avoid the project report as you mentioned earlier. Nevertheless, we think it is better to keep separated the results from the discussion. Thus, the reader can draw more easily conclusions during the discussion.

L616-630: The paragraph for possible multi-resolution approach with AGRIF library may be shortened as it is not part of this study. Fig.10 gives readers a false impression that the 3-level configuration has been tested. It is better removed.
Yes, because this development is still an on-going perspective, we decide to reduce it with just a short mention.

L700: Better not quote conference abstract with a broken web link.
Actually, the link https://ui.adsabs.harvard.edu/abs/2016EGUGA..1815272D is not broken. We have checked again that all the links provided in the bibliography are functional.

**References**

Biastoch, A., Sein, D., Durgadoo, J. V, Wang, Q. and Danilov, S.: Simulating the Agulhas system in global ocean models – nesting vs. multi-resolution unstructured meshes, Ocean Model., 121, 117–131, doi:10.1016/j.ocemod.2017.12.002, 2018.

Chakraborty, S., Arnab, S. and Kambekar, A.: Effect of Climate Change and Sea Level Rise Along the Coastline of Mumbai in 2050-using MIKE 21, J. Offshore Struct. Technol., 8(3), 55–64, doi:10.13140/RG.2.2.18691.78880, 2021.

Debreu, L. and Blayo, E.: Two-way embedding algorithms: a review, Ocean Dyn., 58(5), 415–428, doi:10.1007/s10236-008-0150-9, 2008.

Debreu, L., Marchesiello, P., Penven, P. and Cambon, G.: Two-way nesting in split-explicit ocean models: Algorithms, implementation and validation, Ocean Model., 49–50, 1–21, doi:https://doi.org/10.1016/j.ocemod.2012.03.003, 2012.

Dufois, F., Verney, R., Le Hir, P., Dumas, F. and Charmasson, S.: Impact of winter storms on sediment erosion in the Rhone River prodelta and fate of sediment in the Gulf of Lions (North Western Mediterranean Sea), Cont. Shelf Res., 72, 57–72, doi:https://doi.org/10.1016/j.csr.2013.11.004, 2014.

De Goede, E. D.: Historical overview of 2D and 3D hydrodynamic modelling of shallow water flows in the Netherlands, Ocean Dyn., 70(4), 521–539, doi:10.1007/s10236-019-01336-5, 2020.

Lazure, P. and Dumas, F.: An external–internal mode coupling for a 3D hydrodynamical model for applications at regional scale (MARS), Adv. Water Resour., 31(2), 233–250, doi:10.1016/J.ADVWATRES.2007.06.010, 2008.

Li, J. G.: Filling oceans on a spherical multiple-cell grid, Ocean Model., 157(February 2020), 101729, doi:10.1016/j.ocemod.2020.101729, 2021.

Ménesguen, A., Dussauze, M. and Dumas, F.: Designing optimal scenarios of nutrient loading reduction in a WFD/MSFD perspective by using passive tracers in a biogeochemical-3D model of the English Channel/Bay of Biscay area, Ocean Coast. Manag., 163, 37–53, doi:https://doi.org/10.1016/j.ocecoaman.2018.06.005, 2018.

Muller, H., Blanke, B., Dumas, F., Lekien, F. and Mariette, V.: Estimating the Lagrangian residual circulation in the Iroise Sea, J. Mar. Syst., 78, S17–S36, doi:https://doi.org/10.1016/j.jmarsys.2009.01.008, 2009.

Parsapour-Moghaddam, P. and Rennie, C. D.: Hydrostatic versus nonhydrostatic hydrodynamic modelling of secondary flow in a tortuously meandering river: Application of Delft3D, River Res. Appl., 33(9), 1400–1410, doi:https://doi.org/10.1002/rra.3214, 2017.

Shchepetkin, A. F. and McWilliams, J. C.: The regional oceanic modeling system (ROMS): a split-explicit, free-surface, topography-following-coordinate oceanic model, Ocean Model., 9(4), 347–404, doi:https://doi.org/10.1016/j.ocemod.2004.08.002, 2005.

---

## Author Comment (AC2)

**Response to reviewer 2**

We first want to thank you for paying very careful attention and spending time to review our work; we carefully cope with your remarks and advices that greatly help to improve our manuscript. We hope these modifications make the manuscript significantly clearer and thus worth publishing.

This manuscript describes the implementation of 2-way nesting into the 3D ocean model MARS3D using the AGRIF software package. The model is then applied to several domains to demonstrate its utility, specifically around the issue of residence times in coastal waters.

*Major points:*

The manuscript is generally fairly well written (but with frequent grammatical errors), although it does seem to be conflicted about its focus. It's hard to discern what this is; is it to introduce the implementation of AGRIF into another ocean code, characterise the physics of the Bay of Brest region or further demonstrate the benefits of 2-way nesting. If the former two, then I think the paper falls short. If the latter, then this is not entirely new. The discussion section reads as if the object of the paper is indeed this improvement over 1-way nesting. However, the configuration chosen to make these comparisons is hardly unique, and the 1-way configuration to which the 2-way is compared seems set up to fail. Similar objectives of resolution placement and computational cost may be achieved by exercising the curvilinear capabilities of 1-way grid, for example, a polar-style curvilinear application may similarly achieve high resolution at the coast but decrease resolution at the seaward open boundaries. Such an application may provide equivalent dynamical advantages to the 2-way approach. Also, compared to the continuous resolution transition across scales afforded by unstructured models, the two-way approach presents a rather brutal interface that requires work (e.g., sponge zones, restriction operators) to produce a viable solution. In this sense the unstructured approach is a more elegant and dynamically consistent approach. If benefits of 2-way nesting are the focus, it would be interesting to compare to, or at least consider, a tailored curvilinear grid or unstructured approach at the same resolution in the target area. My guess is the solutions would be similar in all cases if grid/meshes are carefully constructed, so benefits would come down to throughput. This would make for an interesting paper, but probably not within scope of what's tractable here. An important advantage of 1-way nesting is that once the open boundary files have been generated, then they can be re-used at no cost, whereas for 2-way nesting the parent must be also run every time the child is optimised, or an additional experiment performed. Since the child obviously runs faster than the parent+child, in terms of overall throughput this may be a more efficient system if many runs of the child are required. The re-use of child OBC conditions probably deserves consideration. Regardless, if comparative 1/2-way approaches are considered, then a more rigorous consideration of throughput could be included alongside dynamical benefits, and optimized 1-way configurations should be explored.

This first major point is quite exhaustive and a bit hard to address. Nevertheless, we try to argument here below and accounted for most of the points within the manuscript itself.
The fact that you found hard to discern what are the main points of this manuscript reveals that we must re-organize it. The major topic is mostly the implementation of AGRIF refinement capability in the MARS3D model in a two-way mode: after a description of the method, we give two examples of use that illustrate the built-in capability. We deeply reorganize the text in order this point of view is more straightforwardly catchable by the reader. In that sense we agree with the need of clarification pointed out in your review. We try to avoid confusion induced by comparing one way and two-way approach that is clearly not our main goal here.
"Is that very new or not" is a point that is let to the discretion to the reviewer. Our point of view is that it is worth to illustrate an implementation of AGRIF in a split implicit model (MARS) which raises different question with respect to the implementation in a split explicit solver (e.g., ROMS). It is also for us a step forward as long as what was described in the literature was only one-way (Muller et al., 2009) and partially two-way (Dufois et al., 2014). Since then, many improvements were performed that are now described in further details in the scope of the paper.
We also take advantage of your point of view to position our problem in a broader perspective of the community of ocean modeler that have to cope with local / targeted refinement. We enriched the introduction with more references that described a large span of approaches to reach this goal. But as you pointed it out, it is largely beyond the scope of this study to compare or cross-evaluate different approaches. Your arguments are very valuable, there is no doubt and even we do agree with you:
    - We mention and quote some alternative approaches for the local refinement would it be structured (Diaz et al., 2020; Rétif et al., 2014… among many others) or unstructured (Comblen et al., 2010; Guerin et al., 2016; Qi et al., 2018). We also pointed out that the design of curvilinear structured orthogonal or even non orthogonal curvilinear structured grid (Grasso et al., 2018) might be a great deal of work with a lack of flexibility. Without an efficient grid mesh generator, the nesting proposed here keeps it simple for the end-user. Considering unstructured grid, it is recalled it induces a significant computational cost or constrain required by the finest grid cells to not spill over the entire grid. Yet the reviewer #1 mentioned the existence of spherical multiple-cell grid model developed by Li (2021) which circumvents this constraint.
    - We remind the argument in favor of a one-way approach in case one has to perform repeated experiments for tuning purpose or whatever. The comparison made between one-way and two-way nesting methods are based on recent configurations with the same geographic extent and recently used for environmental studies (Cadier et al., 2017; Gangnery et al., 2019; Petton et al., 2020). So, it is still relevant to compare them with the two-way nesting approach. However, you are correct to mention the possibility of using curvilinear grid in order to bypass numerical issues at open boundaries. The one-way nesting stays for now what is typically performed in applications with our partners. The use of AGRIF approach induced a significant computation cost due to the inegration over the whole grid hierarchy. You are absolutely right that the re-use of child OBC computed once for good is a lighter solution in case several runs are performed with different parametrizations or different environmental assumptions. However, this is typically dependent of the final objective. The regional configuration presented was mainly used to produce a hindcast over 10

years. It is still running in operational mode. Finally, the two-way nesting is a solution with MARS3D to insure a conservative approach (biological tracer, connectivity study…) over large geographic area at minimum cost.
Nevertheless, our point is just to promote one among many other relevant solutions keeping in mind that none of these fulfill all the desired requirements. So, we mention in the new release of the paper some of your arguments.
Last, we point out that the physics characterization of the Bay of Brest is not within the scope of this study. It is a case study where the two-way nesting is not necessary for the hydrodynamics inside the bay whereas it is compulsory to obtain a valid time renewal indicator with mass conservation update and forcing between mother and child grid.

The system characterisation of the region comprises validation against tidal observations and T/S. The flushing time is then computed (using 1 and 2-way) for the Bay of Brest. Although the paper considers several flushing metrics, a variation of the non-stationary method (e.g., Tartinville et al., 1997) is used. The authors speculate on the physics responsible for the distribution of these flushing times, but I'm not sure this is a significant addition to previous studies of the region cited in Section 2.3.1
As quoted previously, our goal was not to focus on the system characterization but rather to illustrate pros of the refinement methodology (implemented here with two levels of grids) that copes with the tidal prism oscillations more accurately. Such time scale indicator was not accurately estimated for this bay at this horizontal resolution (Le Pape and Menesguen, 1997; Pommepuy et al., 1979). Using the two-way method is a new way to get validated and spatially detailed flushing metrics for such control volume. The foremost finding obtained is the overestimation of the renewal capacity of the bay when using one-way nesting compared to two-way nesting. The structure distribution is explained by the integration of neap-spring tidal cycle and the runoff intensity. The indicator reliability was accessed using the same model with a smaller control volume; in that case both one-way and two-way nesting estimate the same renewal time.

The application of 2-way nesting is not new, neither is the implementation of AGRIF in coastal modes. In Section 2.2, it seems that AGRIF has previously been included in MARS3D (e.g., Dufois et al), so it seems that the MARS3D+AGRIF combination is not new either (?). One of the challenges of using AGRIF to orchestrate 2-way nesting previously has been the issue of coupling at the barotropic level. Since all ocean models are either mode split, or semi-implicit (as is this one), then to maintain stability and accuracy in 2-way systems it is generally required that information be exchanged between coarse and fine grids at every barotropic step in the case of the former and every iteration of the implicit solver in the latter. I'm unsure if AGRIF has been applied to semi-implicit models, as I believe the implementations in NEMO, ROMS and HYCOM use a mode-split approach. This potentially is a point of difference in this study that should be exploited, requiring better articulation of coupling at the barotropic level.
You are perfectly informed and our purpose is exactly what you pointed out: the originality of the implementation described (now in further details) in the paper is the fact that barotropic solver and the barotropic / baroclinic coupling is very different in MARS3D than in the three other models quoted which all have a split explicit free surface. What is reported here is some steps ahead of what have been used earlier. These past implementations solely assured the mass conservation with one-way nesting (Muller et al., 2009) and in two-way nesting only for the tracers (Dufois et al., 2014). The MARS3D code gathers now numerous developments for fully operational two-way nesting, and the implementation of AGRIF in a split semi-implicit surface model was never performed to our knowledge. We think it is worth presenting it as long as the Alternate Direction Implicit (ADI *ie* semi-implicit) solver used in MARS3D is quite popular among coastal modelling community (Chakraborty et al., 2021; De Goede, 2020; Parsapour-Moghaddam and Rennie, 2017) concerning model as widely spread as Delft3D or Mike21. Considering this remark, the article was restructured to better identify what is new in MARS3D model, AGRIF library and the two-ways nesting technique implementation. The Sect. 2 is now devoted to innovative developments with reference to previously published studies.

***Minor points:***

P1. L15. 'institute for the exploitation', do you mean 'exploration'?
No, exploitation is still the right acronym for our institute created in 1984.

P1, L16. 'preserves some essential principles', suggest replace 'principles' with 'properties'.
Performed.

P1, L16. 'constant preserving', do you mean preserving constancy condition, monotonic, positive definite? What are the 'induced constraints'?
We meant "constancy condition" but we replaced it with "mass and momentum conservations". The induced constraints are mainly the need of bathymetric coherence and the increase of computation cost.

P1, L22. 'the paper intends at comparing', suggest changing to 'intends to compare'.
Performed.

P1, L23. 'how MARS3D-AGRIF tool', suggest 'how the MARS3D-AGRIF tool'. 'efficient way significantly', suggest 'efficient way to significantly'.
Performed.

P1, L24. 'bring it biological issues'. What does this mean?
We replace the sentence by "unravel ecological challenges".

P1, L28. 'and surely for a long time', suggest changing to 'for a long time'.
Done.

P1, L31. 'grid on key-locations', suggest 'grid at key locations'. 'region can varies', suggest 'region can vary'.
Done.

P2, L36. 'solitary waves train', suggest 'solitary wave train'.
Performed.

P2, L38. 'And then they', suggest changing to 'Then they'.
Performed.

P3, L70. 'introduces shortly, suggest changing to 'introduces briefly'.
Thank you for this suggestion, we delete those words as the Sect. 2 has been split in two. Sect. 2 is now dedicated to the innovative developments compare to previous published works. Sect 3 depicted both configurations and time scale indicator.

P3, L83. 'allows to enhance the', suggest 'allows the enhancement of'.
Thank you for this suggestion, we remove this sentence to clarify this subsection to new developments in MARS3D V11.2.

P4, L98. Perhaps for completeness explicitly specify the function G.
Actually, it is explained just after the equations. It gathers the vertical average of all the remaining terms including the non-linear and the dissipation terms, the Coriolis force, the friction at the surface and the bottom.

P4, Equation 2. Perhaps some introduction of eqn. 2 is required to make known its purpose; is this simply an expansion of eqn 1? For what purpose – what are you trying to show here?
The system (2) is rigorously equivalent to the system (1). It is just a reassortment:
- For the sake of readability, we introduce $u^{n+1,*}$
- If we expand $u^{n+1,*}$ into the fourth equation of equations system (2), we get exactly the third equation of the equations system (1)
- If we gather the first term of the right hand side of the third equation of the equations system (2) $(-g\Delta t^2(1-\alpha)\frac{\partial}{\partial x}\left[h^n\frac{\partial\eta^{n+1}}{\partial x}\right])$ with the second one of the left hand side $(\Delta t\left[\frac{\partial}{\partial x}(h^n u^{n+1,*})\right])$ and expand $u^{n+1,*}$ we obtain $\Delta t\left[\frac{\partial}{\partial x}(h^n u^{n+1})\right]$ which is exactly the second term of the left hand side of the second equation of system (1)

This is worth to mention it as long as the system (1) leads to a global solver for a given (here) row that tight $\eta^{n+1}$ and $u^{n+1}$ in the unknown vector whereas the system (2) leads to one local solver (for $u^{n+1}$) and one global solver for the same given row which unknown vector is made solely of $\eta^{n+1}$: it is then twice smaller. It is an efficient way of saving MPI communication as long as the global communications between tiles are reduced for the barotropic solver by a factor of 2.

P4, L115. Perhaps a brief overview of this is required. Either include the full equations or reference Lazure and Dumas and make it clear the above refers to the semi-implicit method (as opposed to explicit mode-split).
We agree with you and we develop this section with previous refence and a clear mention to semi-implicit method.

P6, Eq 5 & 6. These could probably be made clearer by a more explicit formulation.
OK, we only partially addressed this remark as long as there is no clear in-between the full complex development and the formulation as it stood.

P6, L162. Do the iterative strategies used by Haley, Martin mean that 2-way nesting has been previously used in semi-implicit models? If so, this probably needs to be stated earlier.
According to our knowledge it is not the case. Nevertheless as long as this is not a crucial point in our purpose, we suppressed these references and this allusion for the sake of simplicity.

P7, L187-189. What about conservation for the fine grid using interpolated coarse grid variables? Is this interpolation conservative?
Yes, such as the update step, the interpolation is conservative. We add specific details about this important section.

P9, L265. What are the 'classical open boundary conditions'? Sommerfeld radiation (which ones), Flather radiation, adaptive radiation, Dirichlet, upstream advection (characteristics)? The OBC used plays a key role in model accuracy and stability, and some further consideration here is probably warranted. These choices will likely impact the validation metrics in Section 3.2.
You are right, the term "classical" was definitely too obvious considering the wide range of possible schemes. We remove it and specify the scheme used clearly in the manuscript.

P10. L292-293. It'd be good to show the control domain on a diagram.
Thank you for this remark. The western boundary of the control domain was added with a dash line in the Fig. 6.

P10. L298. 'tracer concentration equal to 0'. This is for inflow – what about outflow concentrations?
As the inflow concentration of water coming from the ocean are set to 0, the outflow concentrations are not taking into account. There is no zero gradient condition apply to the boundary of the model. This is only right for the larger model (250m Iroise mother

grid for two-way nesting; 50m Brest grid for one-way nesting). The back-and-forth flow through the control volume border is properly taking into account. We detail more this point.

P10. L299. 'between 95% and 1/e'. Is it 95%, or 1/e (which is ~36%)?
We need to clarify this point. The flushing lag $t_0$ represents the time decrease for the concentration to reach 95%. Then the local flushing time represents the time between this flushing lag (95%) and the time decrease up to $1/e$. On one point of the grid, the global flushing time is the addition of the flushing lag and the local flushing time. Removing the flushing lag is important especially in geographic complex areas. We re-order this paragraph to be clearer.

P11. L303. 'To get rid of the initial conditions', what does this mean? 'the 13 tracers' – what are the 13 tracers? Perhaps clarify.
Plus et al. (2009) demonstrated that the release time has an effect depending on the initial tide moment. Therefore, a release of 13 tracers that covers every hour for the tidal cycle is needed to get a more robust indicator. We delete the word "the" and added the above reference.

P12. L348. Change 'ODDYSEA' to 'ODYSSEA'.
Performed.

P13. L356. 'according to the chosen boundary scheme'. What is the boundary scheme? These details are important.
Yes, you are right. It's a prime interest in this study and we should have given more details about it. We specify explicitly which open boundary scheme was used for currents and tracers in the previous sections for the one-way nesting. As this paragraph concerns the impacts of offline interpolation, the chosen boundary scheme is not relevant here so we rephrase this sentence.

P13, L368. Change 'computational coast' to 'computational cost'.
Performed.

P14, L390. What about comparison to the low frequency component? This is a harder test for the model. The tidal component only really needs to be compared over a neap-spring cycle, whereas the low frequency component requires a much longer series.
Thank you for this suggestion. The PREVIMER tidal components atlas does not provide maps for low frequency waves such as Ssa, Mm or Mf. Any comparison would then be questionable. Though, the difference between one-way and two-way for theses waves are below one centimeter. It is the reason why we did not mention this detail.

P15, Table 4. I'm assuming the amplitude in cm is for the model, and the relative difference is the model-observation difference. If that's the case then some of the K1 metrics don't seem to line up (e.g. an observation of the 1-way amplitude overestimated by 9% doesn't seem commensurate with a 2-way amplitude of 8.8cm overestimated by 29%).
You are right for the quantities and the units; we modify the legend for more clarity. For the example chosen, both nesting methods overestimate the actual value of K1 wave which is 6.82cm. As you well spotted, it is hard to understand these differences without knowing the sign of the relative difference. We add it to the table consequently.

P15, Section 3.3. What are the specific improvements of 2-way nesting? Perhaps use this paragraph to better introduce Sections 3.3.1 and 3.3.2.
We actually reorganize the manuscript by presenting sequentially both realistic configurations. In this way, we hope the improvements of two-way nesting will be clearer for the reader.

P15, L429. At what locations do the Taylor diagrams correspond to? Perhaps mention the gauges in Fig 4 here.
You are right: the comparisons in the Taylor diagrams are related to the gauges presented in Fig. 3 and Fig. 6. We had a specific comment about this.

P16, L447. 'This could be explained by the nesting feedback that enables a more accurate temperature budget in the mother grid'. Perhaps if a high-resolution grid were designed with a polar curvilinear application that maintained high resolution at the coast but pushed the open boundaries further into the area occupied by the mother, then a similar result would be achieved to the 2-way approach. i.e., any difference may be a consequence of the configuration design due to shortcomings of the 1-way approach not being properly accounted for (in this case with open boundaries too close to the area of interest).
Yes, you are right. The shortcomings of the one-way approach are revealed due to the lack of simulated domain compared to the two-way nesting. A far larger grid (with extra computing time) or another kind of grid as you quoted (unstructured, polar curvilinear, …) would have been a fair comparison. However, it is largely beyond the scope of this study to compare or cross-evaluate. We highlight the fact that this one-way approach is not the finest one.

P16, Section 3.3.2. The Ushant front is stated to be due to tidal mixing. Table 4 indicates that, on balance, the 1-way approach has a better representation of the tidal height (although this is mainly due to K1, which appear to have some inconsistencies in their reporting). This comparison to sea level implies the barotropic currents (and hence tidal mixing) are similar (or better) for 1-way compared to 2-way models; i.e., the 2-way exchange of barotropic currents doesn't seem to improve tidal currents in the model interior over 1-way nesting. If this is the case, and resolution is the same in the 1 and 2-way Iroise zoom (500m), then why is the 1-way result worse in terms of the front? Some speculation around the dynamics causing this may benefit.
Not only the tidal mixing is responsible for the Ushant front but also the initial spatial distribution of temperature field (Brumer et al., 2020). The tide is indeed better represented with the one-way nesting because this model is forced with the SHOM CST-France model (112 harmonic components). Nevertheless, this upwind scheme used for tracers at open boundaries is not sufficient for the

heat exchange at an hour frequency. This crucial aspect needed to be better highlighted, as the simulated domain is not large enough to bypass this issue. As you mentioned earlier, a more adequate one-way nesting with polar curvilinear should have overcome it. We describe

P18, L494. 'initially the flushing lag'. What is the flushing lag's value?
We should have said "the initial flushing lag" as it represents the time decrease for the concentration to reach 95%. Here for the control volume, it ranges from few hours next to western border up to 7 days in the eastern part of the bay. We clarify the description of the time indicator and give the range of the flushing lag.

P18. L495. 'The same analysis….', is this analysis the Bay of Brest 50m model forced by the regional 250m model? Perhaps clarify.
Yes, as said in the manuscript, the same simulations are made with the one-way nesting configuration for each scenario. The Brest 50m model is run solely forced by the regional 250m model. We replace the sentence to clarify the approach.

P18, L498. 'The one-way nesting overestimates…', how do you know the 1-way is incorrect and the 2-way is the better estimate?
You are definitely entitled to ask which one is the better. The same approach was used with a far smaller control volume located only inside the bay. As the simulated domain is larger than the control volume, results were the same between nesting methods. We add a specific comment about this.

P18. L513. 'on behalf of various reasons', suggest changing to 'due to various reasons'.
Done.

P18, L515. 'AGRIF library as used for a split free surface ocean model'. Isn't the model semi-implicit rather than mode-split?
Yes, you are absolutely right. It is a mistake and we correct it.

P18, L517. 'requires to write and store the 3D forcing file', suggest 'writing and storage of 3D forcing files'.
Done.

P18, L518. 'grid are picked up.', suggest changing to 'grid are supplied'.
Done.

P19, L519. 'high frequency writings', suggest 'high frequency output'.
Done.

P19, L520. 'other kind of issues', suggest 'other kinds of issues'.
Done.

P19, L521. 'Despite there are many improvements to deal with that question', suggest 'Despite the many improvements to deal with this question,'.
Done.

P19, L523. 'the cost of long-term storage of massive data.' Only data on the open boundaries need be stored, which may not be that massive, even at high frequency. Also, storing data enables reuse for the child only, without having to run both the parent-child.
Yes, only the open boundaries of the child grid need to be stored in one-way nesting. It is the main actual solution used with MARS code but the interpolated OBC data on child grid at high frequency (sometimes 5 mins for keeping momentum and density coherence) could represent a large space. As mentioned earlier, the re-use of child OBC is a lighter solution if several runs are needed with different parametrizations or different environmental hypothesis. However, this is typically depending on the final objective. We add a specific statement concerning this point.

P19, L525. 'sketched' – what does this mean in this context? Perhaps rephrase.
We simplify the sentenced by saying "more flexibility than AGRIF provides".

P19, L526. '… by performing them online at each time step….'. This is true, but there is no possibility of re-use, which may be a disadvantage in some cases.
Yes, you are right, the time computation remains the main drawback of the two-way nesting. For testing numerical developments, the two-way nesting is not a straightforward process. It is more adapted to create hydrodynamic hindcast or to transport tracer along long geographic area with different horizontal resolutions. We discuss more about this drawback in the manuscript.

P19, L530. 'consists in the', suggest changing to 'consists of the'.
Ok

P19, L537. 'constraints prevent from gravity issues', suggest 'constraints prevent the gravity issues'.
Ok

P19, L538. 'the same cares in the grid', suggest 'the same care in defining the grid'.
Ok

P19, L538-543. This argument is not particularly convincing as a burning issue. Perhaps rephrase or omit.
Thank you for this remark. We rephrase this point and reduce it to one sentence.

P19, L545. 'than the one reached with the tidal forcing prescribed at open boundary conditions', suggest changing to ' than the one achieved with the tidal forcing at open boundaries'.
Ok

P19, L547. 'For that kind of standalone grids', suggest 'For this type of standalone grid'.
Ok

P19, L548. 'they enables to represent accurately', suggest 'enabling the accurate representation of'.
Ok

P19, L549. 'thanks to adapted open boundary conditions algorithm', suggest ' via the open boundary condition algorithm'.
Ok

P19, L550. 'performed once for good', suggest 'performed only once'.
Ok

P20, L553-555. This is confusing and doesn't get the point across. Please rephrase. 'They are not straightforward..' – what does this mean?
You are right, this is confusing. We wanted to say the differences were not one way or the other between the both nesting methods. For the sake of clarity, we delete this sentence.

P20, L557. 'maintains MARS3D good ability to', suggest changing to 'allows MARS3D to'.
Ok

P20, L559. 'exposed previously, suggest 'previously presented'.
Ok

P20, L565. 'large scales as the tidal forcing', suggest 'large scales such as the tidal forcing'.
Ok

P20, L570. 'boundary effect', suggest 'boundary effects'.
Ok

P20, L580. Why not make the standalone grid encompass the entire control volume, similar to the parent? Although, this would come at extra computational cost – would that be tolerable?
The standalone grid indeed encompasses the control volume but it should have the geographic extent of the parent grid (Iroise sea). In our case, this would represent an 8-fold size increase (6-fold by taking into account land value). The computation cost would then be multiplied by at least the same factor.

P20, L585. 'tidally flushed of the bay', suggest 'tidally flushed from the bay'.
Ok

P21, L590. 'conservative approach required', suggest 'conservative approach is required'.
Ok

P21, L587-591. A polar coordinate 1-way nest may achieve the same result. Even better would be an unstructured coastal or bathymetry weighted mesh. Perhaps mention the 2-way solution is not unique.
Thank you for this suggestion.

P21, L592. 'the AGRIF library flexibility allows to specify', suggest 'the flexibility of the AGRIF library allows the specification of'.
Ok

P21, L594. 'of the different', suggest 'to the different'.
Ok

P21, L595. There is no reference to the seven-zoom grid apart from its layout. Without evidence it functions this should probably be removed.
You are right, this could be misleading for the reader. We remove the term seven-zoom and let the regional configuration reference.

P21, L592-593. This is an important advantage of 2-way nesting, and should be highlighted up front. Experiments demonstrating this would really bolster the case for 2-way nesting.

Actually, it is already the case for meteorological forcings for the regional configuration. As AROME model does not cover all Europe, it only forced the zoom child grids. The mother grid is forced with ARPEGE model. On a more physical parameter, the horizontal turbulent closure consists of a Laplacian operator with a constant turbulent viscosity coefficient. This coefficient differs between each zoom ranging from 0.5 up to 3 $m^2 s^{-1}$. We highlight this aspect according to your rightful suggestion.

P21, L602. 'without additional task', suggest changing to 'without additional tasks'.
Ok

P21, L605. 'In situation where', suggest 'In situations where'.
Ok

P21, L606. 'checked once offline' – checked for what?
If a child grid is removed, the mother grid's bathymetry of the on the child grid area need to be recomputed in order to assure coherence between bathymetry at a center of a cell and its four borders required by MARS3D especially for wetting and drying scheme. We replace the sentence by saying "the mother grid's bathymetry has just to be recomputed before launching the model".

P21, L610. 'In such a perspective', suggest 'In such a case'.
Ok

P22, L623. 'with a refinement factor of 2 to 5'. Unstructured meshes have continuous resolution transition, and as long as this is sufficiently smooth, any refinement factor can be accommodated.
Yes, this was more a general comment from our bias experience with AGRIF in coastal areas. You are right that the refinement factor is the consequence of the physical and geographical aspects. As we reduce this paragraph, we delete this sentence.

P21, L623-630. If the hierarchy has not been actively exercised, with results to show, then probably best not to include reference to it.
Yes, because this development is still an on-going perspective, we decide to reduce it with just a short mention

P21, L634. 'allows to reach', suggest changing to 'allows us to reach'.
Thank you, the reviewer #1 suggest to change this sentence to "allow reaching" so we follow its advice.

P21, L640. 'to monitor marine environment', suggest 'to monitor the marine environment'.
Ok

P21, L641. 'the MARS3D model give the', suggest 'the MARS3D model provides the'.
Ok

**References**

Brumer, S. E., Garnier, V., Redelsperger, J.-L., Bouin, M.-N., Ardhuin, F. and Accensi, M.: Impacts of surface gravity waves on a tidal front: A coupled model perspective, Ocean Model., 154, 101677, doi:https://doi.org/10.1016/j.ocemod.2020.101677, 2020.

Cadier, M., Gorgues, T., Sourisseau, M., Edwards, C. A., Aumont, O., Marié, L. and Memery, L.: Assessing spatial and temporal variability of phytoplankton communities' composition in the Iroise Sea ecosystem (Brittany, France): A 3D modeling approach. Part 1: Biophysical control over plankton functional types succession and distribution, J. Mar. Syst., 165, 47–68, doi:https://doi.org/10.1016/j.jmarsys.2016.09.009, 2017.

Chakraborty, S., Arnab, S. and Kambekar, A.: Effect of Climate Change and Sea Level Rise Along the Coastline of Mumbai in 2050-using MIKE 21, J. Offshore Struct. Technol., 8(3), 55–64, doi:10.13140/RG.2.2.18691.78880, 2021.

Comblen, R., Blaise, S., Legat, V., Remacle, J.-F., Deleersnijder, E. and Lambrechts, J.: A discontinuous finite element baroclinic marine model on unstructured prismatic meshes, Ocean Dyn., 60(6), 1395–1414, doi:10.1007/s10236-010-0357-4, 2010.

Diaz, M., Grasso, F., Le Hir, P., Sottolichio, A., Caillaud, M. and Thouvenin, B.: Modeling Mud and Sand Transfers Between a Macrotidal Estuary and the Continental Shelf: Influence of the Sediment Transport Parameterization, J. Geophys. Res. Ocean., 125(4), e2019JC015643, doi:https://doi.org/10.1029/2019JC015643, 2020.

Dufois, F., Verney, R., Le Hir, P., Dumas, F. and Charmasson, S.: Impact of winter storms on sediment erosion in the Rhone River prodelta and fate of sediment in the Gulf of Lions (North Western Mediterranean Sea), Cont. Shelf Res., 72, 57–72, doi:https://doi.org/10.1016/j.csr.2013.11.004, 2014.

Gangnery, A., Normand, J., Duval, C., Cugier, P., Grangeré, K., Petton, B., Petton, S., Orvain, F. and Pernet, F.: Connectivities with Shellfish Farms and Channel Rivers are Associated with Mortality Risk in Oysters, Aquac. Environ. Interact., 11, 493–506, doi:10.3354/aei00327, 2019.

De Goede, E. D.: Historical overview of 2D and 3D hydrodynamic modelling of shallow water flows in the Netherlands, Ocean Dyn., 70(4), 521–539, doi:10.1007/s10236-019-01336-5, 2020.

Grasso, F., Verney, R., Le Hir, P., Thouvenin, B., Schulz, E., Kervella, Y., Khojasteh Pour Fard, I., Lemoine, J.-P., Dumas, F. and Garnier, V.: Suspended Sediment Dynamics in the Macrotidal Seine Estuary (France): 1. Numerical Modeling of Turbidity Maximum Dynamics, J. Geophys. Res. Ocean., 123(1), 558–577, doi:https://doi.org/10.1002/2017JC013185, 2018.

Guerin, T., Bertin, X. and Dodet, G.: A numerical scheme for coastal morphodynamic modelling on unstructured grids, Ocean Model., 104, 45–53, doi:https://doi.org/10.1016/j.ocemod.2016.04.009, 2016.

Li, J. G.: Filling oceans on a spherical multiple-cell grid, Ocean Model., 157(February 2020), 101729, doi:10.1016/j.ocemod.2020.101729, 2021.

Muller, H., Blanke, B., Dumas, F., Lekien, F. and Mariette, V.: Estimating the Lagrangian residual circulation in the Iroise Sea, J. Mar. Syst., 78, S17–S36, doi:https://doi.org/10.1016/j.jmarsys.2009.01.008, 2009.

Le Pape, O. and Menesguen, A.: Hydrodynamic prevention of eutrophication in the Bay of Brest (France), a modelling approach, J. Mar. Syst., 12(1–4), 171–186, doi:10.1016/S0924-7963(96)00096-6, 1997.

Parsapour-Moghaddam, P. and Rennie, C. D.: Hydrostatic versus nonhydrostatic hydrodynamic modelling of secondary flow in a tortuously meandering river: Application of Delft3D, River Res. Appl., 33(9), 1400–1410, doi:https://doi.org/10.1002/rra.3214, 2017.

Petton, S., Pouvreau, S. and Dumas, F.: Intensive use of Lagrangian trajectories to quantify coastal area dispersion, , doi:10.1007/s10236-019-01343-6, 2020.

Plus, M., Dumas, F., Stanisière, J. Y. and Maurer, D.: Hydrodynamic characterization of the Arcachon Bay, using model-derived descriptors, Cont. Shelf Res., 29(8), 1008–1013, doi:10.1016/j.csr.2008.12.016, 2009.

Pommepuy, M., Manaud, F., Monbet, Y., Allen, G., Salomon, J. C., Gentien, P. and L'Yavang, J.: ETUDE OCEANOGRAPHIQUE APPLIQUEE AU S.A.U.M. DE LA RADE DE BREST, pp. 15–16., 1979.

Qi, J., Chen, C. and Beardsley, R. C.: FVCOM one-way and two-way nesting using ESMF: Development and validation, Ocean Model., 124(March), 94–110, doi:10.1016/j.ocemod.2018.02.007, 2018.

Rétif, F., Bouchette, F., Marsaleix, P., Liou, J.-Y., Meulé, S., Michaud, H., Lin, L.-C., Hwang, K.-S., Bujan, N., Hwung, H.-H. and Team, S.: REALISTIC SIMULATION OF INSTANTANEOUS NEARSHORE WATER LEVELS DURING TYPHOONS, Coast. Eng. Proc., 1(34 SE-Waves), waves.17, doi:10.9753/icce.v34.waves.17, 2014.

---

## Author Response (AR1)

Dear Qiang Wang,

Following the recommendations of the two reviewers, we have carefully corrected our manuscript in order to greatly improve its comprehension and its relevance.

We hope these modifications make the manuscript significantly clearer and thus worth publishing.

Best regards,
Sébastien Petton

---

## Referee Report (RR1)

Comments on the revised manuscript titled:

**Using two-way nesting technique AGRIF with MARS3D V11.2 to improve hydrodynamics and estimate environmental indicators**

by Sébastien Petto, Valérie Garnie, Matthieu Caillaud, Laurent Debreu and Franck Dumas

The revised manuscript has addressed some of issues mentioned in my first review. Added material to show the parallel computing cost is quite useful and clarifying new development reported in this manuscript is also helpful. Although the authors have provided some arguments in their responses to my first review comments, some issues remain unsolved.

1. The manuscript still looks too long for a journal paper. As the main development is adding the two-way nesting, it should concentrate on this point and minimize other technique details to shorten it. Two-way nesting is not a novel technique. This should be classified as an application to this particular system.

2. Whether the MARS3D-AGRIF system is a convenient tool for coastal marine application is still a question though the authors maintained their claim. From my understanding, the system is complicated to set up and awkward for post-processing, especially when there are many nested child grids. The authors have claimed that they can build a hierarchy of 4 or even more levels to meet coastal refinement but this will make the system even more complicated as they have explained in section 5 discussion. Adding a child grid requires offline bathymetric adjustment, physical parameter tuning, extra care for vertical level alignment, and boundary setup for overlapping with neighbouring child-grids. Considering that there might be over hundreds of child-grids when more levels are added, the system is doomed to be complicated. The authors argued that file compression may reduce the waste of storing land point values but storing output from over hundreds of child-grids already makes the output processing a huge burden. I think this is not the authors fault but the flaw of the system design. It would be unreasonable to ask the authors to simplify the system in a short time. I recommend that the authors just clarify it to avoid misleading readers.

---

## Author Response (AR2)

**Editor**

Dear authors,

Both reviewers made some comments to further improve the manuscript. I agree with their main points. Please take their comments into account to revise the manuscript.

Best wishes

Qiang Wang

Dear Editor,

Following the recommendations of the two reviewers, we modified and shortened by 15% our manuscript in order to highlight its relevance. We hope these modifications make the manuscript significantly clearer and thus worth publishing.

Best regards,

Sébastien Petton

**Report 1:**

The revised manuscript has addressed some of issues mentioned in my first review. Added material to show the parallel computing cost is quite useful and clarifying new development reported in this manuscript is also helpful. Although the authors have provided some arguments in their responses to my first review comments, some issues remain unsolved.

1. The manuscript still looks too long for a journal paper. As the main development is adding the two-way nesting, it should concentrate on this point and minimize other technique details to shorten it. Two-way nesting is not a novel technique. This should be classified as an application to this particular system.

We do concede that the manuscript is quite long compare to the normed. However, it could be explained by the recent addition of the numerical developments realized in MARS and in in order to use full two-ways nesting in MARS model. In addition, we have taken care to properly demonstrate the various improvements. As said in the previous review step, the implementation of AGRIF in a split semi-implicit surface model was never performed to our knowledge. We think it is worth presenting it as long as the Alternate Direction Implicit (ADI *i.e.,* semi-implicit) solver used in MARS3D is quite popular among coastal modelling community. Nevertheless, we shortened the manuscript by 15% to comply with your requirement. We removed any redundant idea, the timescale indicator introduction and the conclusion part.

2. Whether the MARS3D-AGRIF system is a convenient tool for coastal marine application is still a question though the authors maintained their claim. From my understanding, the system is complicated to set up and awkward for post-processing, especially when there are many nested child grids. The authors have claimed that they can build a hierarchy of 4 or even more levels to meet coastal refinement but this will make the system even more complicated as they have explained in section 5 discussion. Adding a child grid requires offline bathymetric adjustment, physical parameter tuning, extra care for vertical level alignment, and boundary setup for overlapping with neighbouring child-grids. Considering that there might be over hundreds of child-grids when more levels are added, the system is doomed to be complicated. The authors argued that file compression may reduce the waste of storing land point values but storing output from over hundreds of child-grids already makes the output processing a huge burden. I think this is not the authors fault but the flaw of the system design. It would be unreasonable to ask the authors to simplify the system in a short time. I recommend that the authors just clarify it to avoid misleading readers.

As you spotted, the building of multi-resolution configuration with hundreds of zoom grids represents a real challenge for next generation of hydrodynamic models. The complicated process of creating this hierarchy (grid generation, overlap area fitting, bathymetric adjustment) is partially achieved at the moment with an external Fortran tool based on AGRIF library. The vertical alignment is automatically realized thanks to the system design. The physical parameter tuning and boundary setup fixing are part of the validation process and cannot be bypassed one way or another, as in any other hydrodynamic model. Upon us, the determining factors are now composed of parallel optimization computation and output files generation with the minimal overlap data. For the latter, we have already two solutions in mind depending on the final study objectives as said in the previous review: the user could choose to get a unique grid per level with all the child grid's data gathered upon weight averaged on overlapping areas and without keeping data where there is a subgrid level; or it will rely on a more classic format like the one used in unstructured model with a set of 1D nodes (longitude, latitude) where only data at the highest resolution will be kept.

This is still a major development and it was mentioned here as future perspectives. Therefore, we decided to remove this last paragraph of the discussion to avoid any misleading information.

**Report 2:**

The authors have implemented significant restructuring of the manuscript to improve its focus. The paper now flows much better, and presents as a study showing improvements to regional dynamics using 2-way nesting as diagnosed by comparison to observation (tides, temperature, salinity) and consideration of a flushing metric. As mentioned in the previous review, such implementations of 2-way nesting are not new, but here would add weight to the case in favour of using 2-way nesting. It's probably the editor's call to decide if this work significantly adds to the pool of literature on the subject such that it is worthy of publication.

The authors have included considerably more detail about how the semi-implicit method is handled in MARS. However, this doesn't translate to a clear picture of how the barotropic mode is coupled between coarse and fine grids. It seems like the ADI solver is not iterative, and computes updated barotropic velocities in one step (e.g., P4, L194). In terms of the 2-way nesting, then the coarse grid cannot supply barotropic velocities to the fine grid at shorter time-steps (as could be done with explicit split models at the barotropic timestep, or semi-implicit models with matrix inversions at every iteration of the implicit solver). If this is the case, then it should be clearly stated in Section 2.2. Also, some statement should be made on stability and accuracy of not coupling coarse and fine grids at a barotropic step, and perhaps speculate why it's not necessary when many split explicit models do require this for stability.

Yes, you are right, the barotropic and baroclinic velocities are computed in a single time step (see equations system 3). Then the coupling between both modes is straightforward (see L115-119). Nevertheless, the coarse grid forces every sub time step of the fine grid, thanks to the spatio-temporal conservative interpolator P (see L149-151). Doing so, the fine grid follows the trend of the coarse one. After the complete integration of the fine grid, the update procedure (from fine to coarse grid) finalizes the coupling. Consequently, the coupling is done at every (half) time step of the mother grid. Half is for a row-column scan for instance).

In the paper, this chronologic procedure is summarized in equations of system 6:
1. The coarse grid evaluates its dynamic over a half time step.
2. Then the coarse grid supplies boundary conditions (sea surface elevation, barotropic and baroclinic velocities, temperature, salinity) to the child grid at shorter time-steps thanks to the spatio-temporal interpolator P (Sect. 2.2.1).
3. After the complete sub-time steps integration, the coarse solution is updated using the child solution

The same integration is realized afterward for the other half time step of the coarse grid for the calculation of $\eta$, $v$, $u$ (column-row-wise, instead of row-column-wise for the first half time step).

We concede that the barotropic fields, which force the child grid, do not come directly from the coarse grid computation (as for explicit split models). But it is worth noticing that the conservative interpolation allows us to introduce several levels of child grids. Following your remark, we specified on L153 and L163 that the described procedure is valid for each half time step of the coarse grid.

These, and the points below, are minor additions to the manuscript to strengthen its case, and I recommend minor revision.

P1, L13. 'As for structured grids', suggest 'for structured grids'.
Ok

P1, L16. 'coastal environmental researches and studies', suggest 'coastal environmental research and studies'.
Ok

P3, L121. 'a regional modelized configuration', suggest 'a regional model configuration'. 'a focus coastal' suggest 'a focussed coastal'.
Ok

P3, Eq. 1. Perhaps state u and v are depth averaged velocities. These are written with an overbar in Fig 2?
Thank you for this remark. For the sake of clarity, we had an overbar in each concerned equation. We also corrected Eq. 4 for the estimation of $f_{wet}$ coefficient.

P6, L. 176. It seems that tracers on the boundary are computed using a radiation boundary condition (should this be characteristics – e.g., upstream advection?), rather than using interpolated coarse grid fluxes to update tracers via divergence. This gives global, rather than local, conservation, which is far less useful. Perhaps acknowledge here that local conservation is not achieved at the boundary.
You are right, the tracers are forced thanks to a radiation boundary condition not based on divergence of heat/salt fluxes. We explicitly specified that local conservation is not achieved at the boundary due to the interpolation. The same remark is valid for the velocities. However, it is not so penalizing for these fields because the momentum equations are not written in flux form. Moreover, at the open boundaries, the velocities are only used in Coriolis and Non-linear terms. Concerning the tracers, the misfit is however reduced thanks to the high coherence between the fine and coarse grids and the conservative interpolation for entering/exiting fields (Piecewise Parabolic Method in the normal direction to the boundary).

Section 2.2.3, first para. It's not clear over which part of the coarse grid the restriction operator is applied. Some 2-way nesting applies the restriction operator to the full coarse grid domain, others only to the interface separation (area of overlap). The last sentence of this paragraph says split explicit models apply R to 'a limited updated area next to the boundary of overlapping', whereas MARS uses R on 'the full area of overlapping'. This isn't clear; does the former mean the interface separation plus some extra area beyond the overlapping, and the latter only overlapping? Please clarify.
This note about split explicit models was in fact too general. For such models, the restriction operator could only be applied to a few meshes on the inner contour of the interface between mother and child grids. For MARS model, the restriction must be applied to the whole overlapping area. We clarify this point by only mentioning what is realized in our case.

P7, L197. 'has be obtained' suggest 'has been obtained'.
Done

P8, L217. Here it's acknowledged that global mass conservation is achieved. Perhaps also acknowledge that the more stringent requirement of local mass conservation is not.
We explicitly wrote in the paper that the update procedure is fully conservative (thanks to the procedure itself and the perfect match between the child and mother grid bathymetries) and that only the global conservation is achieved (due to potential misfits from the interpolation). The non-local conservation is specified in the previous section (L171).

P8, L221. 'at same hierarchical grid level', suggest 'at the same hierarchical grid levels'.
Done

P8, L225. It's not clear how the weights are computed – is it the distance to its own open boundary, or the boundary of the child grid it overlaps?
Actually, it is a mix of both criteria you cited: The weights are defined on the overlapping area according to an order of priority with respect to a limit of mother cells inside the child grid at its interface (three times the spatial refinement factor). In this limit, the child grid does not have priority, the weights are decreasing reaching 0 at its boundaries. Outside this limit, the daughter grid takes priority and the weights are defined according to the position of the border of the other overlapping child grid. We clarified this point from L220 to L226.

P9, L255. Change 'nesting ang' to 'nesting and'.
Ok

P11, L306. 'A zero gradient condition is applied to currents'. Is this both normal and tangential currents. What's the boundary condition for elevation (Sommerfeld radiation, Flather etc)?
Yes, it is both for normal and tangential currents. A clamped boundary condition is set for the elevation with FES model. We added these specifications to the manuscript. L290-291

P14, L365. 'depth is only of 8 m', suggest 'depth is only 8 m'.
Ok

P22, L898. 'relies more on the accuracy of available bathymetric data than technical issues' – is this really true? How much effort is required to optimize restriction operators, interface separation and sponge zones in order to achieve a viable solution? Is the system really insensitive to these operators?
Technically, creating an additional zoom is a process that only requires three steps (creation of zoom grid with respect to mother, interpolation of bathymetric data and connection with grid hierarchy). In practice, one has also to consider validation step, and of course this requires paying attention to interface dynamics. The key element lies in the geographic extent of the zoom depending also on the bathymetric gradient. As this assumption is only partially true, and to comply with Reviewer #1 we remove the conclusion and this sentence.

P22, L901. 'that require mass conservation' – BGC typically requires local mass conservation; i.e. no mass is created or destroyed locally (e.g. at the coarse-fine boundary). Perhaps clarify this.
You are right, we should have been more specific and we clarified this in the paper (see above answers). Regarding the conclusion, we removed it (and thus this sentence) to comply with Reviewer #1.